# FlowRL: Matching Reward Distributions for LLM Reasoning

**Xuekai Zhu[1]  Daixuan Cheng[6]  Dinghuai Zhang[3]  Hengli Li[5]  Kaiyan Zhang[4,9]**
**Che Jiang[4]  Youbang Sun[4]  Ermo Hua[4]  Yuxin Zuo[4]  Xingtai Lv[4]**
**Qizheng Zhang[7]  Lin Chen[1]  Fanghao Shao[1]  Bo Xue[1]  Yunchong Song[2]**
**Zhenjie Yang[1]  Ganqu Cui[2]  Ning Ding[4,2]  Jianfeng Gao[3]  Xiaodong Liu[3]**
**Bowen Zhou[4,2,*]  Hongyuan Mei[8,*]  Zhouhan Lin[1,2,*]**

[1]Shanghai Jiao Tong University   [2]Shanghai AI Laboratory   [3]Microsoft Research
[4]Tsinghua University   [5]Peking University   [6]Renmin University of China
[7]Stanford University   [8]Toyota Technological Institute at Chicago   [9] Frontis.AI

`hongyuanmei@gmail.com`   `xuekaizhu0@gmail.com`   ⭕ FlowRL
*Corresponding Authors.

## Abstract

We propose FlowRL: matching the full reward distribution via flow balancing instead of solely maximizing rewards in large language model (LLM) reinforcement learning (RL). Recent advanced reasoning LLMs adopt reward-maximizing methods (*e.g.*, PPO and GRPO), which tend to over-optimize dominant reward signals while neglecting less frequent but valid reasoning paths, thus reducing diversity. In contrast, we transform scalar rewards into a normalized target distribution using a learnable partition function, and then minimize the reverse KL divergence between the policy and the target distribution. We implement this idea as a flow-balanced optimization method that promotes diverse exploration and generalizable reasoning trajectories. We conduct experiments on both math and code reasoning tasks: FlowRL achieves a significant average improvement of $10.0\%$ over GRPO and $5.1\%$ over PPO on math benchmarks, and performs consistently better on code reasoning tasks. These results highlight reward distribution-matching as a key step toward efficient exploration and diverse reasoning of LLM reinforcement learning.

## 1 Introduction

Reinforcement learning (RL) plays a crucial role in the post-training of large language models (LLMs) (Zhang et al., 2025b). A series of powerful reasoning models (Guo et al., 2025; Kavukcuoglu, 2025; Rastogi et al., 2025) have employed large-scale reinforcement learning to achieve strong performance on highly challenging benchmarks. The evolution of RL algorithms for LLM reasoning has progressed through several key stages: REINFORCE (Sutton et al., 1999a) provides a solid baseline that is easy to implement and efficient in simple settings; PPO (Schulman et al., 2017) improves upon REINFORCE with better stability and efficiency in complex settings; GRPO (Shao et al., 2024) simplifies PPO training by eliminating the learning of a separate value function and relying on group comparisons. However, all these methods share a fundamental limitation in their reward-maximizing objective.

Reward-maximizing RL methods tend to overfit to the dominant mode of the reward distribution (Skalse et al., 2022; Pan et al., 2022; Zelikman et al., 2022; Gao et al., 2023). As illustrated in Figure 1, representative RL methods such as GRPO neglect other meaningful modes, which often results in limited diversity among generated reasoning paths and reduces generalization to less frequent yet valid logical outcomes (Hu et al., 2023). These drawbacks become especially pronounced in complex long-chain-of-thought (CoT; Wei et al., 2022) reasoning, where capturing a diverse distribution of plausible solutions is essential for effective generalization (Liu et al., 2025a). Recent approaches adjust the clip ratio (Yu et al., 2025b), apply entropy-based advantage shaping (Cheng

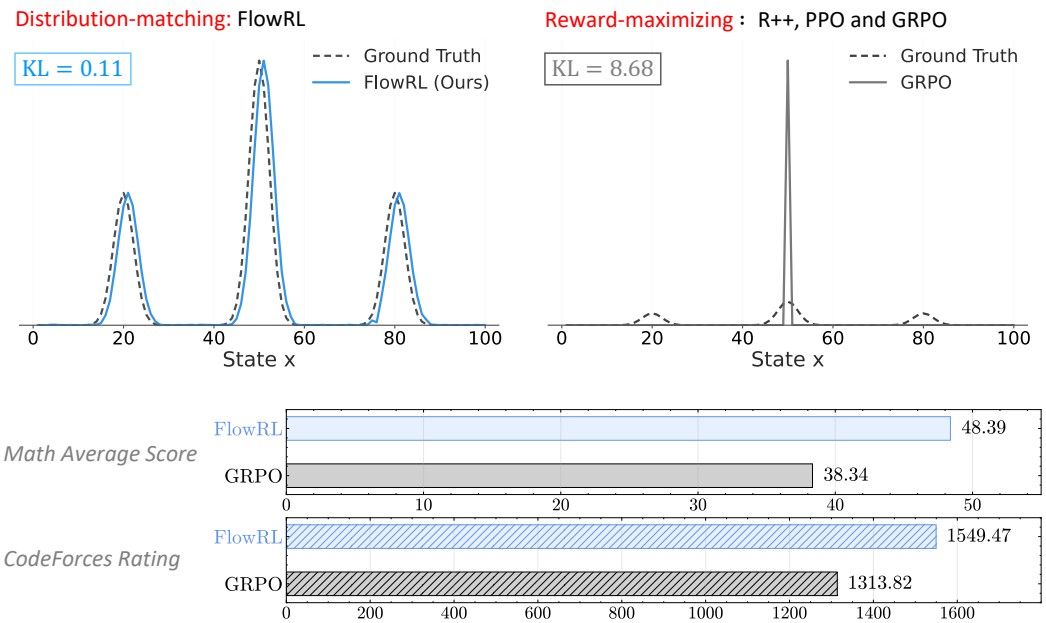

Figure 1: **Top**: Comparison between distribution-matching and reward-maximizing approaches. FlowRL (left) learns to match the full reward distribution, maintaining diversity across multiple modes with low KL divergence. In contrast, reward-maximizing methods (right) such as RE-INFORCE++ (R++; Sutton et al., 1999b; Hu et al., 2025), PPO (Schulman et al., 2017), and GRPO (Shao et al., 2024) concentrate on a single high-reward peak, leading to mode collapse and higher KL divergence. **Bottom**: Performance comparison. FlowRL consistently outperforms GRPO across math and code domains.

et al., 2025), or selectively promote high-entropy tokens (Wang et al., 2025), thereby dynamically adapting the data distribution and implicitly increasing diversity. This raises a fundamental question: How can we promote diverse exploration to prevent convergence to dominant solution patterns in RL training?

In this paper, we propose **FlowRL**, a policy optimization algorithm that aligns the policy model with the full reward distribution, encouraging mode coverage. FlowRL achieves more efficient exploration by fundamentally shifting from reward maximization to reward distribution matching, thereby addressing the inherent mode-collapse limitations of previous RL approaches. As illustrated in Figure 1, the core idea of FlowRL is to introduce a learnable partition function that normalizes scalar rewards into a target distribution, and to minimize the reverse KL divergence between the policy and this reward-induced distribution. We develop this KL objective based on the trajectory balance formulation from GFlowNets (Bengio et al., 2023b), providing a gradient equivalence proof that bridges generative modeling and policy optimization. To address the challenges of long CoT training, we introduce two key technical solutions: *length normalization* to tackle gradient explosion issues that occur with variable-length CoT reasoning, and *importance sampling* to correct for the distribution mismatch between generated rollouts and the current policy.

We compare FlowRL with mainstream RL algorithms for LLM reasoning, including REIN-FORCE++ (Hu et al., 2025), PPO, and GRPO, across math and code domains, using both base and distilled LLMs with 7B or 32B parameters. In the math domain, FlowRL outperforms GRPO and PPO by 10.0% and 5.1%, respectively, demonstrating consistent improvements on six challenging math benchmarks (MAA, 2025; 2023; Lightman et al., 2023a; Lewkowycz et al., 2022; He et al., 2024). Furthermore, FlowRL surpasses both PPO and GRPO on three challenging coding benchmarks (Jain et al., 2024; Penedo et al., 2025; Chen et al., 2021), highlighting its strong generalization capabilities in code reasoning tasks. To understand what drives these performance gains, we analyze the diversity of generated reasoning paths and confirm that FlowRL produces substantially more diverse rollouts than the baseline methods, thereby validating the effectiveness of our approach in exploring multiple solution strategies.

**Contributions.** We summarize the key contributions of this work as follows:

- We propose FlowRL, a policy optimization algorithm that shifts from reward maximization to reward distribution matching via flow balancing, encouraging diverse reasoning path exploration while addressing the inherent mode-collapse limitations of existing RL methods.
- We introduce length normalization and importance sampling to enable effective training on variable-length CoT reasoning, addressing gradient explosion and sampling mismatch issues.
- FlowRL outperforms GRPO and PPO by 10.0% and 5.1% respectively across math benchmarks and demonstrates strong generalization on code reasoning tasks, with diversity analysis confirming substantially more diverse solution exploration.

## 2 PRELIMINARIES

**Reinforcement Learning for Reasoning.** We formulate reasoning as a conditional generation problem, where the policy model receives a question $\mathbf{x} \in \mathcal{X}$ and generates an answer $\mathbf{y} \in \mathcal{Y}$. The objective is to learn a policy $\pi_\theta(\mathbf{y}|\mathbf{x})$ that produces high-quality answers under task-specific reward signals $r$. To better illustrate the policy optimization procedure, we provide a detailed formulation of GRPO below. For each question $\mathbf{x}$, GRPO samples a group of answers $\{\mathbf{y}_1, \mathbf{y}_2, \dots, \mathbf{y}_G\}$ from old policy $\pi_{\theta_{old}}$ and updates the model by maximizing the following objective:

$$\mathcal{J}_{GRPO}(\theta) = \mathbb{E}_{[\mathbf{x}\sim P(\mathcal{X}),\{\mathbf{y}_i\}_{i=1}^{G}\sim\pi_{\theta_{old}}(\mathcal{Y}|\mathbf{x})]}$$

$$\frac{1}{G}\sum_{i=1}^{G}\frac{1}{|\mathbf{y}_i|}\sum_{t=1}^{|\mathbf{y}_i|}\left\{\min\left[\frac{\pi_\theta(\mathbf{y}_{i,t}|\mathbf{x},\mathbf{y}_{i,<t})}{\pi_{\theta_{old}}(\mathbf{y}_{i,t}|\mathbf{x},\mathbf{y}_{i,<t})}\hat{A}_{i,t}, \text{clip}\left(\frac{\pi_\theta(\mathbf{y}_{i,t}|\mathbf{x},\mathbf{y}_{i,<t})}{\pi_{\theta_{old}}(\mathbf{y}_{i,t}|\mathbf{x},\mathbf{y}_{i,<t})}, 1-\epsilon, 1+\epsilon\right)\hat{A}_{i,t}\right] - \lambda\mathbb{D}_{KL}\left[\pi_\theta||\pi_{ref}\right]\right\},$$

$$\mathbb{D}_{KL}(\pi_\theta||\pi_{\text{ref}}) = \frac{\pi_{\text{ref}}(\mathbf{y}_i|\mathbf{x})}{\pi_\theta(\mathbf{y}_i|\mathbf{x})} - \log\frac{\pi_{\text{ref}}(\mathbf{y}_i|\mathbf{x})}{\pi_\theta(\mathbf{y}_i|\mathbf{x})} - 1,$$

(1)

where $\epsilon$ and $\lambda$ are hyper-parameters. Here, $A_i$ denotes the advantage, computed by normalizing the group reward values $\{r_1, r_2, \dots, r_G\}$ as $A_i = \frac{r_i - \text{mean}(\{r_1, r_2, \cdots, r_G\})}{\text{std}(\{r_1, r_2, \cdots, r_G\})}$. Compared to GRPO, REINFORCE applies the policy gradient directly, without advantage normalization, clipping, or KL regularization. PPO uses a critic model to estimate the advantage and employs importance sampling to stabilize policy updates.

**GFlowNets.** Generative Flow Networks (GFlowNets; Bengio et al., 2023a) are a probabilistic framework for training stochastic policies to sample discrete, compositional objects (*e.g.*, graphs, sequences) in proportion to a given reward. As shown in Figure 2, the core principle of GFlowNets is to balance the forward and backward probability flows at each state, inspired by flow matching (Bengio et al., 2021). The initial flow is estimated by $Z_\phi(s_0)$ at the initial state $s_0$. The output flow is equal to the outcome reward $r(s_f)$ conditioned at the final state $s_f$. Following Lee et al. (2024), we use a 3-layer MLP to parameterize $Z_\phi$. This flow-balancing mechanism facilitates the discovery of diverse, high-reward solutions by ensuring proper exploration of the solution space. See Appendix C for detailed GFlowNets background.

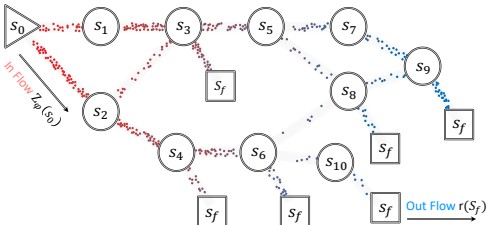

Figure 2: GFlowNets (Bengio et al., 2023a), a flow-balance perspective on reinforcement learning. The initial flow $Z_\phi(s_0)$ injects probability mass into the environment, which is transported through intermediate states by the policy $\pi_\theta$ and accumulated at terminal states in proportion to the scalar rewards.

## 3 METHODOLOGY

In this section, we first formulate distribution matching in reinforcement learning through reverse KL divergence and establish its connection to trajectory balance from GFlowNets. To address the challenges of gradient explosion and sampling mismatch encountered during long CoT training, we

further incorporate length normalization and importance sampling. Using this enhanced framework, we derive a flow-balanced objective, termed *FlowRL*.

## 3.1 FROM REWARD MAXIMIZATION TO DISTRIBUTION MATCHING

As illustrated in Figure 1, recent powerful large reasoning models typically employ reward-maximizing RL algorithms, such as PPO or GRPO. However, these methods tend to optimize toward the dominant reward mode, frequently resulting in mode collapse and the neglect of other plausible, high-quality reasoning paths. To address this fundamental limitation, we propose optimizing the policy by aligning its output distribution to a target reward distribution. A simple yet effective way to achieve this is to minimize the reverse KL divergence[1] between the policy and this target. However, in long CoT reasoning tasks, the available supervision in RL is a scalar reward, rather than a full distribution. Moreover, enumerating or sampling all valid trajectories to recover the true reward distribution is computationally intractable.

Inspired by energy-based modeling (Hinton et al., 1995; Du & Mordatch, 2019), we introduce a learnable partition function $Z_\phi(\mathbf{x})$ to normalize scalar rewards into a valid target distribution. This allows us to minimize the reverse KL divergence between the policy and the reward-weighted distribution, formalized as:

$$\min_\theta \mathcal{D}_{\mathrm{KL}}\left(\pi_\theta(\mathbf{y}\mid\mathbf{x}) \,\middle\|\, \frac{\exp(\beta r(\mathbf{x},\mathbf{y}))}{Z_\phi(\mathbf{x})}\right) \quad\Rightarrow\quad \pi_\theta(\mathbf{y}\mid\mathbf{x}) \propto \exp(\beta r(\mathbf{x},\mathbf{y})), \qquad (2)$$

where $r(\mathbf{x},\mathbf{y})$ is the reward function, $\beta$ is a hyperparameter, $Z_\phi(\mathbf{x})$ is the learned partition function, and the resulting target distribution is defined as $\tilde{\pi}(\mathbf{y}\mid\mathbf{x}) = \frac{\exp(\beta r(\mathbf{x},\mathbf{y}))}{Z_\phi(\mathbf{x})}$. This objective encourages the policy to sample diverse, high-reward trajectories in proportion to their rewards, rather than collapsing to dominant modes as in standard reward maximization.

While the KL-based formulation provides a principled target distribution, we derive a more practical, RL-style objective that facilitates efficient policy optimization.

**Proposition 1.** *In terms of expected gradients, minimizing the KL objective in Eq. 2 is equivalent to minimizing the trajectory balance loss used in GFlowNet (Malkin et al., 2022; 2023; Lee et al., 2024; Bartoldson et al., 2025):*

$$\min_\theta \mathcal{D}_{\mathrm{KL}}\left(\pi_\theta(\mathbf{y}\mid\mathbf{x}) \,\middle\|\, \frac{\exp(\beta r(\mathbf{x},\mathbf{y}))}{Z_\phi(\mathbf{x})}\right) \quad\Longleftrightarrow\quad \min_\theta \underbrace{\left(\log Z_\phi(\mathbf{x}) + \log \pi_\theta(\mathbf{y}\mid\mathbf{x}) - \beta r(\mathbf{x},\mathbf{y})\right)^2}_{\text{Trajectory Balance}}$$

$$(3)$$

**Remark 2** (*Trajectory balance as a practical surrogate for KL minimization*)**.** Given the equivalence established in Proposition 1, the KL-based distribution matching objective can be reformulated as the trajectory balance loss. This reformulation provides a practical optimization approach by using a stable squared loss form rather than direct KL optimization, and by treating $Z_\phi(\mathbf{x})$ as a learnable parameter rather than requiring explicit computation of the intractable partition function. The trajectory balance objective thus serves as a tractable surrogate for reward-guided KL minimization that can be directly integrated into existing RL frameworks.

## 3.2 FLOWRL

As established in Proposition 1, the target reward distribution can be approximated by optimizing the trajectory balance objective. However, applying this objective directly to long CoT reasoning introduces two key challenges:

**Problem I: Exploding gradients from long trajectories.** Trajectory balance is a sequence-level objective, and applying it to long CoT reasoning with up to 8K tokens leads to exploding gradients and unstable updates. This issue is not observed in prior GFlowNets works, which typically operate on short trajectories in small discrete spaces. Specifically, the log-probability term $\log \pi_\theta(\mathbf{y}\mid\mathbf{x})$ decomposes into a token-wise sum, $\sum_t \log \pi_\theta(y_t\mid y_{<t}, \mathbf{x})$, causing the gradient norm to potentially scale with sequence length.

---

[1]We use reverse KL since we can only sample from the policy model, not the target reward distribution.

**Problem II: Sampling mismatch.** Mainstream RL algorithms such as PPO and GRPO commonly perform micro-batch updates and reuse trajectories collected from an old policy $\pi_{\theta_{\text{old}}}$, enabling data-efficient training. In contrast, the KL-based trajectory balance objective assumes fully on-policy sampling, where responses are drawn from the current policy. This mismatch poses practical limitations when integrating trajectory balance into existing RL pipelines.

These limitations motivate our reformulation that retains the benefits of distribution matching while addressing key practical challenges. To enable this reformulation, we first redefine the reward function following established practices in GFlowNets literature (Lee et al., 2024; Bartoldson et al., 2025; Yu et al., 2025a) by incorporating a reference model as a prior constraint on the reward distribution. Specifically, we modify the original $\exp(\beta r(\mathbf{x}, \mathbf{y}))$ to include the reference model:

$$\exp\left(\beta \ r(\mathbf{x}, \mathbf{y})\right) \cdot \pi_{\text{ref}}(\mathbf{y} \mid \mathbf{x}), \tag{4}$$

where $r(\mathbf{x}, \mathbf{y})$ denotes the outcome reward commonly used in reinforcement learning and $\pi_{\text{ref}}$ is the initial pre-trained model. We follow Guo et al. (2025) to use outcome-based reward signals, and apply group normalization to $r(\mathbf{x}, \mathbf{y})$ as $\hat{r}_i = (r_i - \text{mean}(\mathbf{r}))/\text{std}(\mathbf{r})$, where $\mathbf{r} = \{r_1, r_2, \ldots, r_G\}$ denotes the set of rewards within a sampled group. By substituting the redefined reward formulation Eq. 4 into Eq. 3, we derive the following objective[2]:

$$\min_{\theta} \left(\log Z_\phi(\mathbf{x}) + \log \pi_\theta(\mathbf{y} \mid \mathbf{x}) - \beta \ \hat{r}_i(\mathbf{x}, \mathbf{y}) - \log \pi_{\text{ref}}(\mathbf{y} \mid \mathbf{x})\right)^2 \tag{5}$$

**Remark 3** (*Reward shaping via length normalization*)**.** Trajectory balance treats both the initial flow and the outcome reward as sequence-level quantities. In contrast, standard policy optimization methods such as PPO or GRPO assign rewards at the token level and compute gradients at each step. However, for trajectories of varying lengths (*e.g.*, CoT responses), this mismatch can cause the log-probability term $\log \pi_\theta(\mathbf{y} \mid \mathbf{x}) = \sum_{t=1}^{|\mathbf{y}|} \log \pi_\theta(y_t \mid y_{<t}, \mathbf{x})$ to scale with sequence length. To address this, we apply a form of reward shaping by normalizing log-probabilities with respect to sequence length. Specifically, we rescale the term as $\frac{1}{|\mathbf{y}|} \log \pi_\theta(\mathbf{y} \mid \mathbf{x})$, balancing the contributions of long and short sequences and stabilizing the learning signal.

**Remark 4** (*Importance sampling for data-efficient training*)**.** To mitigate sampling mismatch, we employ importance sampling inspired by PPO to stabilize policy updates with off-policy data. We re-weight stale trajectories using the importance ratio $w = \pi_\theta(\mathbf{y} \mid \mathbf{x})/\pi_{\text{old}}(\mathbf{y} \mid \mathbf{x})$, which serves as a coefficient in the surrogate loss. Since our objective focuses on optimizing trajectory balance rather than expected return, we detach the gradient from the current policy to prevent excessive policy drift: $w = \text{detach}[\pi_\theta(\mathbf{y} \mid \mathbf{x})]/\pi_{\text{old}}(\mathbf{y} \mid \mathbf{x})$. For additional stability, we incorporate PPO-style clipping to bound the importance weights: $w = \text{clip}\left(\frac{\pi_\theta(\mathbf{y}|\mathbf{x})}{\pi_{\text{old}}(\mathbf{y}|\mathbf{x})}, 1 - \epsilon, 1 + \epsilon\right)^{\text{detach}}$.

Incorporating these improvements into Eq. 5, we arrive at the following FlowRL objective:

**FlowRL**

$$\mathcal{L}_{\text{FlowRL}} = w \cdot \left(\log Z_\phi(\mathbf{x}) + \frac{1}{|\mathbf{y}|} \log \pi_\theta(\mathbf{y} \mid \mathbf{x}) - \beta \hat{r}(\mathbf{x}, \mathbf{y}) - \frac{1}{|\mathbf{y}|} \log \pi_{\text{ref}}(\mathbf{y} \mid \mathbf{x})\right)^2 \tag{6}$$

where the clipped importance weight $w$ and normalized reward $\hat{r}(\mathbf{x}, \mathbf{y})$ are defined as:

$$w = \text{clip}(\frac{\pi_\theta(\mathbf{y} \mid \mathbf{x})}{\pi_{\text{old}}(\mathbf{y} \mid \mathbf{x})}, 1 - \epsilon, 1 + \epsilon)^{\text{detach}}, \quad \hat{r}_i = \frac{r_i - \text{mean}(\mathbf{r})}{\text{std}(\mathbf{r})}. \tag{7}$$

We use this objective to update the policy parameters $\theta$ during training, and refer to this strategy as *FlowRL*. Implementation details and theoretical analysis are provided in § 4 and § B, respectively.

---

[2]The substitution replaces $\beta r(\mathbf{x}, \mathbf{y})$ in trajectory balance objective Eq. 3 with $\beta r(\mathbf{x}, \mathbf{y}) + \log \pi_{\text{ref}}(\mathbf{y} \mid \mathbf{x})$ to incorporate the reference model constraint.

Table 1: **Results on math reasoning benchmarks.** We report Avg@16 accuracy with relative improvements shown as subscripts. Positive gains are shown in green and negative changes in red. FlowRL outperforms all baselines across both 7B and 32B model scales.

| Models | AIME24 | AIME25 | AMC23 | MATH500 | Minerva | Olympiad | Avg |
|---|---|---|---|---|---|---|---|
| *Qwen2.5-32B-Base, Max Response Len = 8K tokens* | | | | | | | |
| Backbone | 4.58 | 2.08 | 28.59 | 52.48 | 26.99 | 21.37 | 22.68 |
| R++ | $14.79_{+10.21}$ | $9.17_{+7.08}$ | $52.65_{+24.06}$ | $44.35_{-8.13}$ | $17.37_{-9.62}$ | $24.52_{+3.15}$ | 27.14 |
| PPO | $26.87_{+22.29}$ | $20.41_{+18.33}$ | $76.40_{+47.81}$ | $69.17_{+16.69}$ | $28.79_{+1.80}$ | $37.90_{+16.53}$ | 43.25 |
| GRPO | $23.12_{+18.54}$ | $14.58_{+12.50}$ | $76.87_{+48.28}$ | $61.60_{+9.12}$ | $18.95_{-8.04}$ | $34.94_{+13.57}$ | 38.34 |
| FlowRL | $23.95_{+19.37}$ | $21.87_{+19.79}$ | $73.75_{+45.16}$ | $80.75_{+28.27}$ | $38.21_{+11.22}$ | $51.83_{+30.46}$ | **48.39** |
| *Qwen2.5-7B-Base, Max Response Len = 8K tokens* | | | | | | | |
| Backbone | 4.38 | 2.08 | 30.78 | 54.47 | 22.38 | 24.03 | 23.02 |
| R++ | $11.04_{+6.66}$ | $5.41_{+3.33}$ | $66.71_{+35.93}$ | $54.25_{-0.22}$ | $24.37_{+1.99}$ | $27.33_{+3.30}$ | 31.52 |
| PPO | $9.38_{+5.00}$ | $7.29_{+5.21}$ | $63.43_{+32.65}$ | $57.98_{+3.51}$ | $26.53_{+4.15}$ | $27.25_{+3.22}$ | 31.98 |
| GRPO | $13.54_{+9.16}$ | $9.79_{+7.71}$ | $64.53_{+33.75}$ | $57.05_{+2.58}$ | $23.06_{+0.68}$ | $26.88_{+2.85}$ | 32.48 |
| FlowRL | $15.41_{+11.03}$ | $10.83_{+8.75}$ | $54.53_{+23.75}$ | $66.96_{+12.49}$ | $31.41_{+9.03}$ | $34.61_{+10.58}$ | **35.63** |

## 4 EXPERIMENT SETTINGS

**Backbone Models.** There are two learnable modules in Eq. 6: the policy model $\pi_\theta$ and the partition function $Z_\phi$. For the policy model $\pi_\theta$, we use Qwen-2.5-7B/32B (Team, 2024) for math tasks and DeepSeek-R1-Distill-Qwen-7B (DeepSeek-AI, 2025) for code tasks, respectively. The reference model $\pi_{\text{ref}}$ is the corresponding fixed pretrained model. For partition function $Z_\phi$, following Lee et al. (2024), we use a randomly initialized 3-layer MLP with hidden dimensions matching those of the base model. The input to $Z_\phi$ is the mean of the language model's hidden states after encoding the input x, and the output is a scalar value. We detail the implementation of $Z_\phi$ in § F. All training scripts are based on the veRL (Sheng et al., 2024). For the reward function, following Lee et al. (2024), we set the hyperparameter $\beta = 15$.

**Baselines.** We compare our method against three representative reward-maximization RL baselines: REINFORCE++ (R++; Sutton et al., 1999b; Hu et al., 2025), PPO (Schulman et al., 2017), and GRPO (Shao et al., 2024). All baselines follow the official veRL recipes, with consistent training configurations. For fair comparison, all methods use the same learning rate, batch size, and training steps, and are evaluated at convergence using identical step counts.

**Training Configuration.** We experiment on both math and code domains. For the math domain, we use the training set collected from DAPO (Yu et al., 2025b). For the code domain, we follow the setup of DeepCoder (Luo et al., 2025), using their training set. For 7B model training, we use a single node equipped with 8 NVIDIA H800 GPUs (80GB memory each). For 32B model training, we scale to 4 nodes with 32 GPUs to accommodate the larger memory requirements. All experiments use max_prompt_length = 2048 and max_response_length = 8192 across both model sizes. We use a batch size of 512 for math reasoning tasks and 64 for code reasoning tasks. We set the learning rate to 1e-6 and enable dynamic batch sizing in veRL for efficient training. For GRPO and FlowRL, we configure rollout_n = 8, meaning each prompt generates 8 response rollouts as the group size.

**Evaluation Configuration.** For the math domain, we evaluate on six challenging benchmarks: AIME 2024/2025 (MAA, 2025), AMC 2023 (MAA, 2023), MATH-500 (Lightman et al., 2023a), Minerva (Lewkowycz et al., 2022), and Olympiad (He et al., 2024). For the code domain, we evaluate on LiveCodeBench (Jain et al., 2024), CodeForces (Penedo et al., 2025), and HumanEval+ (Chen et al., 2021). For all evaluation datasets, we perform 16 rollouts and report the average Pass@1 accuracy, denoted as Avg@16. We further report rating and percentile for Codeforces. During generation, we use sampling parameters of temperature = 0.6 and top_p = 0.95 for all evaluations. The response length for evaluation is set to 8,192 tokens, consistent with the training configuration.

Table 2: **Results on code benchmarks.** We report metrics with relative improvements shown as subscripts. Positive gains are shown in green and negative changes in red. FlowRL achieves the strongest performance across all three benchmarks.

| Models | LiveCodeBench | | CodeForces | | HumanEval+ |
|---|---|---|---|---|---|
| | Avg@16 | Pass@16 | Rating | Percentile | Avg@16 |
| DeepSeek-R1-Distill-Qwen-7B, Max Response Len = 8K tokens | | | | | |
| Backbone | 30.68 | 49.46 | 886.68 | 19.4% | 80.90 |
| R++ | $30.46_{-0.22}$ | $52.68_{+3.22}$ | $1208.03_{+321.35}$ | $56.8\%_{+37.4\%}$ | $76.61_{-4.29}$ |
| PPO | $35.10_{+4.42}$ | $54.48_{+5.02}$ | $1403.07_{+516.39}$ | $73.7\%_{+54.3\%}$ | $82.32_{+1.42}$ |
| GRPO | $32.75_{+2.07}$ | $52.32_{+2.86}$ | $1313.82_{+427.14}$ | $67.1\%_{+47.7\%}$ | $80.13_{-0.77}$ |
| FlowRL | $\mathbf{37.43}_{+6.75}$ | $\mathbf{56.27}_{+6.81}$ | $\mathbf{1549.47}_{+662.79}$ | $\mathbf{83.3\%}_{+63.9\%}$ | $\mathbf{83.28}_{+2.38}$ |

Table 3: Ablation study on FlowRL with Qwen2.5-7B as the base model. Avg@16 accuracy is reported across six math reasoning benchmarks. IS denotes importance sampling.

| Method | AIME 2024 | AIME 2025 | AMC 2023 | MATH-500 | Minerva | Olympiad | Avg |
|---|---|---|---|---|---|---|---|
| FlowRL | 15.41 | 10.83 | 54.53 | 66.96 | 31.41 | 34.61 | 35.63 |
| w/o IS | 6.25 | 7.91 | 41.40 | 56.97 | 22.19 | 25.52 | 26.71 |
| Zhang et al. (2025a) | 10.41 | 6.66 | 53.75 | 66.50 | 30.97 | 33.72 | 33.67 |

## 5 RESULTS

**Main Results.** Our experimental results, summarized in Table 1 and Table 2, demonstrate that FlowRL consistently outperforms all reward-maximization baselines across both math and code reasoning domains. Table 1 reports results on math reasoning benchmarks using both 7B and 32B base models, while Table 2 presents the corresponding results on code reasoning tasks. On math reasoning tasks, FlowRL achieves the highest average accuracy of 35.6% with the 7B model and 48.4% with the 32B model, surpassing PPO by 5.1% and GRPO by 10.1% on the 32B model. FlowRL shows strong improvements on challenging benchmarks like MATH-500 and Olympiad problems, demonstrating consistent gains across diverse mathematical domains. On code generation tasks, FlowRL achieves compelling improvements with the highest Avg@16 score of 37.43% on LiveCodeBench, a Codeforces rating of 1549.47 with 83.3% percentile ranking, and 83.28% accuracy on HumanEval+, outperforming all baselines across the board. These consistent performance gains across both domains and model scales provide strong empirical evidence that FlowRL's flow-balanced optimization successfully enhances generalization. This improvement comes from promoting diverse solution exploration compared to previous reward-maximizing RL approaches.

**Ablation Studies.** We conduct ablation studies on importance sampling and the $\beta$ hyperparameter. For importance sampling, we compared the performance with and without it, and implemented a combined loss approach proposed by Zhang et al. (2025a) that simultaneously optimizes both GFlowNets and PPO objectives. This combined loss focuses on optimizing diffusion models, and we adapt it to long CoT reasoning tasks for comparison. Table 3 demonstrates that importance sampling substantially improves FlowRL performance across all math reasoning benchmarks. Compared to Zhang et al. (2025a), using importance sampling as a trajectory-level ratio is more suitable than the combined loss of GFlowNets and PPO. The performance drop without importance sampling (from 35.63% to 26.71%) highlights the critical role of correcting for distribution mismatch between rollout generation and policy training. For the hyperparameter $\beta$, we conduct

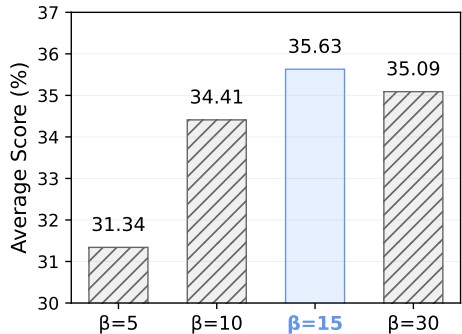

Figure 3: Ablation study on the $\beta$ in FlowRL. $\beta = 15$ (highlighted in blue) achieves the best performance.

a series of parameter ablation studies, and Figure 3 shows that $\beta = 15$ achieves optimal performance, with detailed results shown in Table 7.

## 6  ANALYSIS

**Diversity Analysis.**  To assess solution diversity, we follow the approach of Yu et al. (2025a) and employ `GPT-4o-mini` (OpenAI, 2024) to evaluate all responses generated by each method on AIME 24/25. The evaluation prompt is shown in Appendix H. As shown in Figure 4, FlowRL achieves higher diversity scores compared to baseline methods. This demonstrates that FlowRL improves sample diversity compared to baselines, which tend to exhibit repetitive solution patterns. This diversity evaluation reveals significant differences in exploration patterns across methods. This nearly doubling of diversity score compared to the strongest baseline (PPO) indicates that FlowRL generates qualitatively different solution approaches rather than minor variations of the same strategy. The diversity analysis provides empirical validation of our core hypothesis that flow-balanced optimization promotes mode coverage in complex reasoning tasks.

**Case Study.**  Table 4 illustrates the behavioral differences between GRPO and FlowRL on a representative AIME problem. GRPO exhibits repetitive patterns, applying AM-GM three times and getting stuck in identity loops, failing to solve the problem. FlowRL explores more diverse actions: it sets $a = b$, derives a cubic equation, finds the rational root, and reaches the correct answer. This shows that FlowRL successfully avoids the repetitive exploration patterns. The contrast reveals fundamental differences in exploration strategies: GRPO's reward-maximizing approach leads to exploitation of familiar techniques (AM-GM inequality) without exploring alternatives, eventually reaching contradictory conclusions like $a = b = c$. In contrast, FlowRL's distribution-matching enables strategic decisions such as the symmetry assumption $a = b$, which transforms the problem into a tractable cubic equation $a^3 - 27a + 46 = 0$, allowing systematic solution through rational root testing and polynomial factorization.

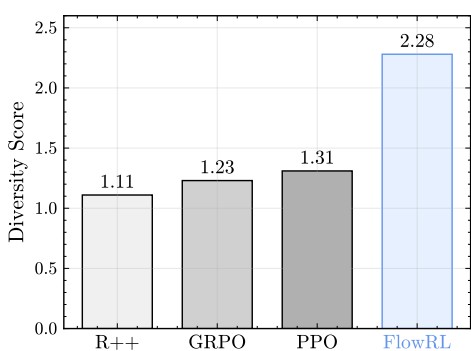

Figure 4: GPT-judged diversity scores on rollouts of AIME 24/25 problems. FlowRL generates more diverse solutions than R++, GRPO, and PPO.

## 7  RELATED WORK

Our work relates to GFlowNets, Flow-Matching Policies, Length Normalization and KL Regularization. We discuss three topics that relate most closely to our work in this section, and the other topics are included in Appendix E.

**Reinforcement Learning for LLM Reasoning.**  RL has emerged as a powerful approach for LLM post-training on reasoning tasks (Sutton et al., 1999b; Schulman et al., 2017; Lightman et al., 2023b; Shao et al., 2024; Guo et al., 2025). Most approaches employ reward-maximizing RL to optimize expected cumulative returns. Entropy regularization (Haarnoja et al., 2018; Ahmed et al., 2019; Cheng et al., 2025) is a classical technique for mitigating mode collapse by promoting diversity in the policy's output distribution, and has also been shown to enhance reasoning capabilities in various settings (Eysenbach & Levine, 2021; Chao et al., 2024). However, for long CoT reasoning, the extended trajectory length (e.g., more than 8k tokens) makes it difficult for the regularization signal to effectively influence reward-maximizing learning. Recent work (Cheng et al., 2025; Wang et al., 2025; Cui et al., 2025; Dong et al., 2025) has discovered that training with more diverse or high-entropy training data can further enhance training effectiveness. Compared to traditional entropy regularization, the above methods explicitly increase the proportion of low-probability (i.e., high-entropy) tokens in the training data. In our work, we address the mode-collapse problem by

Table 4: Case study comparing GRPO and FlowRL rollouts on an AIME problem. GRPO exhibits repetitive patterns (AM-GM ×3, identity loops ×2), while FlowRL follows a more diverse solution path.

| | Content (boxed = actions; "×$k$" = repeated; "…" = omitted) |
|---|---|
| **Question** | Let $\mathcal{B}$ be the set of rectangular boxes with surface area 54 and volume 23. Let $r$ be the radius of the smallest sphere that can contain each box in $\mathcal{B}$. If $r^2 = \frac{p}{q}$ with $\gcd(p, q) = 1$, find $p + q$. |
| **GRPO** | "… denote $a, b, c$ … $\boxed{2(ab+bc+ca) = 54,\, abc = 23}$ … $\boxed{d = \sqrt{a^2 + b^2 + c^2},\, r = d/2}$ … $\boxed{(a+b+c)^2 = a^2+b^2+c^2 + 2(ab+bc+ca)}$ … $\boxed{\text{AM–GM}}$ ×3: $\boxed{\text{AM–GM (1)}}$ … $\boxed{\text{AM–GM (2)}}$ … $\boxed{\text{AM–GM (3)}}$ … $\boxed{(a+b+c)^3 \text{ identity loop}}$ ×2: $\boxed{\text{loop (1)}}$ … $\boxed{\text{loop (2)}}$ … $\boxed{a = b = c \text{ (contradiction)}}$ … $\boxed{\text{back to } (a+b+c)^2}$ … no factorization …" |
| **FlowRL** | "… let $a, b, c$ with $\boxed{2(ab+bc+ca) = 54,\, abc = 23}$ … $\boxed{d = \sqrt{a^2 + b^2 + c^2},\, r = d/2}$ … $\boxed{(a+b+c)^2 \Rightarrow a^2+b^2+c^2 = s^2 - 54}$ … $\boxed{a = b}$ … $\boxed{a^3 - 27a + 46 = 0}$ … $\boxed{\text{rational root } a = 2}$ … $\boxed{\text{factor } (a - 2)(a^2 + 2a - 23)}$ … $\boxed{\text{branch } a = -1 + 2\sqrt{6}}$ … $\boxed{\text{back-sub } c = 23/a^2}$ … $\boxed{a^2+b^2+c^2 = \frac{657}{16}}$ … $\boxed{r^2 = \frac{657}{64}}$ … $\boxed{\text{Answer 721}}$ …" |

fundamentally shifting from reward maximization to reward distribution matching in our RL formulation. See Appendix E for detailed comparisons.

**GFlowNets.** GFlowNets (Bengio et al., 2023a) represent a class of diversity-driven algorithms designed to balance probability flows across states. They have rich connections to probabilistic modeling methods (Zhang et al., 2022a;b; 2024a; Zimmermann et al., 2022; Malkin et al., 2023; Ma et al.), and control methods (Pan et al., 2023b;c;d; Zhang et al., 2024b; Tiapkin et al., 2024). This advantage has enabled GFlowNets to achieve successful applications in multiple downstream tasks, such as molecular drug discovery (Jain et al., 2022; 2023b; Liu et al., 2022; Jain et al., 2023a; Shen et al., 2023; Pan et al., 2023a; Kim et al., 2023; 2024), phylogenetic inference (Zhou et al., 2024), and combinatorial optimization (Zhang et al., 2023a;b). For generative AI, GFlowNets provide a powerful approach to align pretrained models in scenarios such as image generation (Zhang et al., 2025a; Yun et al., 2025) and language model fine-tuning (Hu et al., 2024; Yu et al., 2025a; Lee et al., 2024). Another line of work primarily focuses on the theoretical aspects of GFlowNets. Recent theoretical studies have interpreted GFlowNets as solving a maximum entropy reinforcement learning problem within a modified Markov Decision Process (MDP) (Tiapkin et al., 2024; Deleu et al., 2024; Mohammadpour et al., 2024). These theoretical contributions have inspired us to enhance reinforcement learning from a more foundational standpoint using GFlowNets principles. A comprehensive overview of GFlowNets theory can be found in Appendix C.

**Flow-Matching Policies.** Flow matching simplifies diffusion-based approaches by learning vector fields that transport samples from prior to target distributions (Lipman et al., 2023). Recent work has explored flow matching for policy optimization. McAllister et al. (2025) reformulates policy optimization using advantage-weighted ratios from conditional flow matching loss, enabling flow-based policy training without expensive likelihood computations. Pfrommer et al. (2025) explored reward-weighted flow matching for improving policies beyond demonstration performance. Park et al. (2025) uses a separate one-step policy to avoid unstable backpropagation through time when training flow policies with RL. Zhang et al. (2025a) proposed a combined loss function integrating PPO and GFlowNets to optimize diffusion model alignment. Lv et al. (2025) integrates flow-based policy representation with Wasserstein regularized optimization for online reinforcement learning. However, these approaches focus on continuous control, image generation, or vision-action models, rather than addressing mode-collapse limitations in reward-maximizing RL. Inspired by flow match-

ing principles, our work improves upon RL training to enhance training stability while promoting diverse solution exploration.

## 8 CONCLUSION

In this work, we introduce FlowRL, which transforms scalar rewards into normalized target distributions using a learnable partition function and minimizes the reverse KL divergence between the policy and target distribution. We demonstrate that this approach is theoretically equivalent to trajectory balance objectives from GFlowNets and implicitly maximizes both reward and entropy, thereby promoting diverse reasoning trajectories. To further address gradient explosion and sampling mismatch issues in long CoT reasoning, we incorporate importance sampling and length normalization. Through experiments on math and code reasoning benchmarks, FlowRL achieves consistent improvements across all tasks compared to GRPO and PPO. Our diversity analysis and case studies confirm that FlowRL generates more varied solution approaches while avoiding repetitive patterns.

## ACKNOWLEDGEMENT

This work is sponsored by the National Key Research and Development Program of China (No. 2023ZD0121402) and the National Natural Science Foundation of China (NSFC) grant (No. 62576211). We are grateful to Mingqian Feng and Yuetai Li for their valuable discussions and feedback, which helped improve the quality of this work.

## ETHICS STATEMENT

This work presents FlowRL, a reinforcement learning algorithm for improving reasoning in large language models. Our focus on mathematical and logical problem-solving directly supports beneficial applications in education, scientific research, and decision-support systems. We use established public benchmarks to ensure transparent and unbiased evaluation, and minimize computational waste through efficient configurations, demonstrating our commitment to environmentally conscious and reproducible research.

## REPRODUCIBILITY STATEMENT

We provide comprehensive details to ensure reproducibility: implementation specifics in Section 4 (model architectures, training configurations, hyperparameters), complete algorithmic formulation in Eq. 6, experimental setup covering datasets and evaluation benchmarks, baseline implementations following official veRL recipes, and evaluation methodology. All mathematical formulations, implementation details, and experimental configurations necessary for reproduction are included in the paper.

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

## A    PROOF OF PROPOSITION 1

We begin by analyzing the gradient of the Kullback–Leibler (KL) divergence between the policy $\pi_\theta(\mathbf{y} \mid \mathbf{x})$ and the target reward distribution $\frac{\exp(\beta r(\mathbf{x},\mathbf{y}))}{Z_\phi(\mathbf{x})}$:

$$
\begin{aligned}
&\nabla_\theta D_{\mathrm{KL}}\left(\pi_\theta(\mathbf{y} \mid \mathbf{x}) \,\|\, \frac{\exp(\beta r(\mathbf{x},\mathbf{y}))}{Z_\phi(\mathbf{x})}\right) \\
&= \nabla_\theta \int \pi_\theta(\mathbf{y} \mid \mathbf{x}) \log\left[\frac{\pi_\theta(\mathbf{y} \mid \mathbf{x}) \cdot Z_\phi(\mathbf{x})}{\exp(\beta r(\mathbf{x},\mathbf{y}))}\right] d\mathbf{y} \\
&= \int \nabla_\theta \pi_\theta(\mathbf{y} \mid \mathbf{x}) \log\left[\frac{Z_\phi(\mathbf{x})\pi_\theta(\mathbf{y} \mid \mathbf{x})}{\exp(\beta r(\mathbf{x},\mathbf{y}))}\right] d\mathbf{y} + \int \pi_\theta(\mathbf{y} \mid \mathbf{x}) \nabla_\theta \log\left[\frac{Z_\phi(\mathbf{x})\pi_\theta(\mathbf{y} \mid \mathbf{x})}{\exp(\beta r(\mathbf{x},\mathbf{y}))}\right] d\mathbf{y} \\
&= \int \pi_\theta(\mathbf{y} \mid \mathbf{x}) \nabla_\theta \log \pi_\theta(\mathbf{y} \mid \mathbf{x}) \log\left[\frac{Z_\phi(\mathbf{x})\pi_\theta(\mathbf{y} \mid \mathbf{x})}{\exp(\beta r(\mathbf{x},\mathbf{y}))}\right] d\mathbf{y} + \underbrace{\int \pi_\theta(\mathbf{y} \mid \mathbf{x}) \nabla_\theta \log \pi_\theta(\mathbf{y} \mid \mathbf{x}) \, d\mathbf{y}}_{=\nabla_\theta \int \pi_\theta(\mathbf{y}|\mathbf{x}) \, d\mathbf{y} = \nabla_\theta 1 = 0} \\
&= \int \pi_\theta(\mathbf{y} \mid \mathbf{x}) \nabla_\theta \log \pi_\theta(\mathbf{y} \mid \mathbf{x}) \log\left[\frac{Z_\phi(\mathbf{x})\pi_\theta(\mathbf{y} \mid \mathbf{x})}{\exp(\beta r(\mathbf{x},\mathbf{y}))}\right] d\mathbf{y} \\
&= \mathbb{E}_{\mathbf{y} \sim \pi_\theta(\cdot|\mathbf{x})}\left[\log\left(\frac{Z_\phi(\mathbf{x})\pi_\theta(\mathbf{y} \mid \mathbf{x})}{\exp(\beta r(\mathbf{x},\mathbf{y}))}\right) \cdot \nabla_\theta \log \pi_\theta(\mathbf{y} \mid \mathbf{x})\right]
\end{aligned}
\tag{8}
$$

Next, consider the trajectory balance objective used in GFlowNets learning (Bengio et al., 2023b; Lee et al., 2024; Bartoldson et al., 2025), defined as:

$$
\mathcal{L}(\mathbf{y}, \mathbf{x}; \theta) = \left(\log \frac{Z_\phi(\mathbf{x}) \, \pi_\theta(\mathbf{y} \mid \mathbf{x})}{\exp(\beta r(\mathbf{x},\mathbf{y}))}\right)^2.
\tag{9}
$$

Taking the gradient of this objective with respect to $\theta$ yields:

$$
\nabla_\theta \mathcal{L}(\theta) = 2 \cdot \mathbb{E}_{\mathbf{y} \sim \pi_\theta(\cdot|\mathbf{x})}\left[\left(\log \frac{Z_\phi(\mathbf{x}) \cdot \pi_\theta(\mathbf{y} \mid \mathbf{x})}{\exp(\beta r(\mathbf{x},\mathbf{y}))}\right) \cdot \nabla_\theta \log \pi_\theta(\mathbf{y} \mid \mathbf{x})\right]
\tag{10}
$$

Thus, minimizing the KL divergence is equivalent (up to a constant) to minimizing the trajectory balance loss, confirming Proposition 1.

## B    THEORETICAL ANALYSIS

We conduct an interpretation of FlowRL that clarifies the role of each component in the objective.

**Proposition 5.** *Minimizing the KL divergence in Eq. 5 is equivalent (in terms of gradients) to jointly maximizing reward and policy entropy:*

$$
\max_\theta \, \mathbb{E}_{\mathbf{y} \sim \pi_\theta}\left[\underbrace{\beta \, r(\mathbf{x},\mathbf{y})}_{reward} - \log Z_\phi(\mathbf{x}) + \log \pi_{\mathrm{ref}}(\mathbf{y}|\mathbf{x})\right] + \underbrace{\mathcal{H}(\pi_\theta)}_{entropy}.
\tag{11}
$$

**Remark 6** (*FlowRL beyond reward maximization*). Proposition 5 reveals that FlowRL can be interpreted as jointly maximizing expected reward and policy entropy. This formulation encourages the policy to explore a broader set of high-quality solutions, enabling more diverse and generalizable behaviors on reasoning tasks. Our interpretation also aligns with prior work that views GFlowNets training as a form of maximum entropy RL (Mohammadpour et al., 2024; Deleu et al., 2024).

The proof of Proposition 5 is provided as below.

Recall from Eq. 3 and Eq. 5 that the FlowRL objective is sourced from the minimization of a KL divergence:

$$
D_{\mathrm{KL}}\left(\pi_\theta(\mathbf{y} \mid \mathbf{x}) \,\|\, \frac{\exp(\beta \, r(\mathbf{x},\mathbf{y})) \cdot \pi_{\mathrm{ref}}(\mathbf{y} \mid \mathbf{x})}{Z_\phi(\mathbf{x})}\right) = \int \pi_\theta(\mathbf{y} \mid \mathbf{x}) \log\left[\frac{Z_\phi(\mathbf{x})\pi_\theta(\mathbf{y} \mid \mathbf{x})}{\exp(\beta \, r(\mathbf{x},\mathbf{y})) \cdot \pi_{\mathrm{ref}}(\mathbf{y} \mid \mathbf{x})}\right] d\mathbf{y}
\tag{12}
$$

Rearranging the terms, we obtain:

$$
\begin{aligned}
&\arg\min_{\theta} D_{\mathrm{KL}}\left(\pi_\theta(\mathbf{y}\mid\mathbf{x}) \,\|\, \frac{\exp\left(\beta\, r(\mathbf{x},\mathbf{y})\right)\cdot\pi_{\mathrm{ref}}(\mathbf{y}\mid\mathbf{x})}{Z_\phi(\mathbf{x})}\right) \\
&= \arg\min_{\theta} \int \pi_\theta(\mathbf{y}\mid\mathbf{x})\log\left[\frac{Z_\phi(\mathbf{x})\pi_\theta(\mathbf{y}\mid\mathbf{x})}{\exp\left(\beta\, r(\mathbf{x},\mathbf{y})\right)\cdot\pi_{\mathrm{ref}}(\mathbf{y}\mid\mathbf{x})}\right]d\mathbf{y} \\
&= \arg\max_{\theta}\left\{\mathbb{E}_{\mathbf{y}\sim\pi_\theta(\cdot\mid\mathbf{x})}\log\left[\frac{\exp\left(\beta\, r(\mathbf{x},\mathbf{y})\right)\cdot\pi_{\mathrm{ref}}(\mathbf{y}\mid\mathbf{x})}{Z_\phi(\mathbf{x})}\right] - \int \pi_\theta(\mathbf{y}\mid\mathbf{x})\log\pi_\theta(\mathbf{y}\mid\mathbf{x})d\mathbf{y}\right\} \\
&= \arg\max_{\theta}\left\{\mathbb{E}_{\mathbf{y}\sim\pi_\theta(\cdot\mid\mathbf{x})}\log\left[\frac{\exp\left(\beta\, r(\mathbf{x},\mathbf{y})\right)\cdot\pi_{\mathrm{ref}}(\mathbf{y}\mid\mathbf{x})}{Z_\phi(\mathbf{x})}\right] + \mathcal{H}(\pi_\theta)\right\}
\end{aligned}
\tag{13}
$$

Finally, we express the FlowRL objective in its compact form:

$$
\max_{\theta}\ \mathbb{E}_{\mathbf{y}\sim\pi_\theta(\cdot\mid\mathbf{x})}\left[\underbrace{\beta r(\mathbf{x},\mathbf{y})}_{\text{reward}} - \underbrace{\log Z_\phi(\mathbf{x})}_{\text{normalization}} + \underbrace{\log\pi_{\mathrm{ref}}(\mathbf{y}\mid\mathbf{x})}_{\text{reference model constraint}}\right] + \underbrace{\mathcal{H}(\pi_\theta)}_{\text{entropy}}.
\tag{14}
$$

Therefore, minimizing the FlowRL objective can be interpreted as jointly maximizing reward and entropy, while also aligning the policy with a structured prior. The reward term drives task performance, while the normalization term $Z_\phi(\mathbf{x})$ ensures consistency with a properly normalized target distribution. This encourages the policy $\pi_\theta$ to cover the entire reward-weighted distribution rather than collapsing to a few high-reward modes. The reference policy $\pi_{\mathrm{ref}}$ provides inductive bias that regularizes the policy toward desirable structures, and the entropy term $\mathcal{H}(\pi_\theta)$ encourages diversity in sampled solutions. Together, these components promote better generalization of FlowRL.

## C  GFLOWNETS

We follow the notation of (Madan et al., 2023; He et al., 2025) to introduce the fundamentals of GFlowNets. Let $\mathcal{X}$ denote the compositional objects and $R$ be a reward function that assigns non-negative values to each object $x \in \mathcal{X}$. GFlowNets aim to learn a sequential, constructive sampling policy $\pi$ that generates objects $x$ with probabilities proportional to their rewards, i.e., $\pi(x) \propto R(x)$. This process can be represented as a directed acyclic graph (DAG) $\mathcal{G} = (\mathcal{S}, \mathcal{A})$, where the vertices $s \in \mathcal{S}$ are referred to as *states*, and the directed edges $(u \to v) \in \mathcal{A}$ are called *actions*. The generation of an object $x \in \mathcal{X}$ corresponds to a complete trajectory $\tau = (s_0 \to \cdots \to s_n) \in \mathcal{T}$ within the DAG, beginning at the initial state $s_0$ and ending at a terminal state $s_n \in \mathcal{X}$. The state flow $F(s)$ is defined as a non-negative weight assigned to each state $s \in \mathcal{S}$. The forward policy $P_F(s' \mid s)$ specifies the transition probability to a child state $s'$, while the backward policy $P_B(s \mid s')$ specifies the transition probability to a parent state $s$. To this end, detailed balance objective enforces local flow consistency across every edge $(s \to s') \in \mathcal{A}$:

$$
\forall (s \to s') \in \mathcal{A}, \quad F_\theta(s)P_F(s' \mid s;\theta) = F_\theta(s')P_B(s \mid s';\theta).
\tag{15}
$$

To achieve this flow consistency, GFlowNets employ training objectives at different levels of granularity, including detailed balance (Bengio et al., 2023b), trajectory balance (Malkin et al., 2022), and sub-trajectory balance (Madan et al., 2023). Leveraging their diversity-seeking behavior, GFlowNets have been successfully applied across a range of domains, including molecule generation (Cretu et al., 2024), diffusion fine-tuning (Liu et al., 2025b; Zhang et al., 2025a), and amortized reasoning (Hu et al., 2024; Yu et al., 2025a). Among various training objective in GFlowNets, trajectory balance maintains flow consistency at the trajectory level, defined as:

$$
Z_\theta \prod_{t=1}^{n} P_F(s_t \mid s_{t-1};\theta) = R(x) \prod_{t=1}^{n} P_B(s_{t-1} \mid s_t;\theta).
\tag{16}
$$

Furthermore, sub-trajectory balance achieves local balance on arbitrary subpaths $\tau_{i:j} = \{s_i \to \cdots \to s_j\}$, offering a more stable and less biased learning signal. We build on trajectory balance to extend our KL-based objective through a gradient-equivalence formulation (Prop. 1), and further improve it to better support long CoT reasoning in RL.

Table 5: Math reasoning performance (Avg@64) at temperature $= 0.6$. Relative improvements are shown as subscripts, with positive gains in green and negative changes in red. FlowRL consistently outperforms all baselines and achieves the best average score under this low-temperature setting.

| Models | AIME 2024 | AIME 2025 | AMC 2023 | MATH-500 | Minerva | Olympiad | Avg |
|---|---|---|---|---|---|---|---|
| | | | Qwen2.5-7B Base Model | | | | |
| Backbone | 4.37 | 2.08 | 30.78 | 54.48 | 22.38 | 24.02 | 23.02 |
| R++ | $10.57_{+6.20}$ | $5.10_{+3.02}$ | $66.02_{+35.24}$ | $54.29_{-0.19}$ | $24.47_{+2.09}$ | $27.30_{+3.28}$ | 31.29 |
| PPO | $9.95_{+5.58}$ | $7.34_{+5.26}$ | $63.63_{+32.85}$ | $57.72_{+3.24}$ | $26.22_{+3.84}$ | $27.35_{+3.33}$ | 32.03 |
| GRPO | $14.01_{+9.64}$ | $10.73_{+8.65}$ | $64.10_{+33.32}$ | $57.41_{+2.93}$ | $23.17_{+0.79}$ | $27.11_{+3.09}$ | 32.76 |
| FlowRL | $14.32_{+9.95}$ | $10.05_{+7.97}$ | $55.08_{+24.30}$ | $66.78_{+12.30}$ | $31.52_{+9.14}$ | $34.60_{+10.58}$ | **35.39** |

Table 6: Math reasoning performance (Avg@64) at temperature $= 1.0$. Relative improvements are shown as subscripts, with positive gains in green. FlowRL maintains robust performance under higher generation randomness and continues to outperform all baselines on average.

| Models | AIME 2024 | AIME 2025 | AMC 2023 | MATH-500 | Minerva | Olympiad | Avg |
|---|---|---|---|---|---|---|---|
| | | | Qwen2.5-7B Base Model | | | | |
| Backbone | 3.39 | 1.51 | 23.90 | 45.18 | 16.98 | 18.27 | 18.20 |
| R++ | $10.63_{+7.24}$ | $4.63_{+3.12}$ | $66.99_{+43.09}$ | $54.36_{+9.18}$ | $23.89_{+6.91}$ | $26.65_{+8.38}$ | 31.19 |
| PPO | $10.52_{+7.13}$ | $6.51_{+5.00}$ | $63.04_{+39.14}$ | $57.46_{+12.28}$ | $25.91_{+8.93}$ | $27.16_{+8.89}$ | 31.77 |
| GRPO | $12.50_{+9.11}$ | $10.10_{+8.59}$ | $64.72_{+40.82}$ | $57.15_{+11.97}$ | $23.28_{+6.30}$ | $26.90_{+8.63}$ | 32.44 |
| FlowRL | $14.22_{+10.83}$ | $9.58_{+8.07}$ | $52.92_{+29.02}$ | $66.20_{+21.02}$ | $30.32_{+13.34}$ | $34.47_{+16.20}$ | **34.62** |

Table 7: Ablation study on the effect of the $\beta$ parameter in FlowRL. We report Avg@16 accuracy across six math reasoning benchmarks for different values of $\beta$.

| Models | AIME 2024 | AIME 2025 | AMC 2023 | MATH-500 | Minerva | Olympiad | Avg |
|---|---|---|---|---|---|---|---|
| $\beta = 5$ | 13.54 | 10.00 | 56.09 | 58.91 | 20.79 | 28.72 | 31.34 |
| $\beta = 10$ | 14.79 | 10.20 | 59.53 | 64.30 | 25.27 | 32.39 | 34.41 |
| $\beta = 15$ | 15.41 | 10.83 | 54.53 | 66.96 | 31.41 | 34.61 | 35.63 |
| $\beta = 30$ | 15.00 | 10.83 | 50.62 | 69.02 | 30.03 | 35.03 | 35.09 |

## D  HUMAN STUDY AND CROSS-DOMAIN EVALUATION

**Human Evaluation.**  We conduct a comprehensive human evaluation that demonstrates strong agreement with GPT-4o-mini assessments. We use the same rollouts from the GPT-4o-mini diversity experiment (Sec 6) to further validate diversity. As shown in Table 8, both evaluators independently identify FlowRL as the most diverse method and R++ as the least diverse, with GRPO and PPO showing intermediate diversity levels.

Human Instruction: As a human evaluator, assess the diversity of solutions for each problem by examining 16 solution attempts per method. Rate diversity on a 1-3 scale based on the following criteria:

- Score 1 (low diversity): 13+ responses use essentially identical approaches with only trivial differences in arithmetic, notation, or wording.
- Score 2 (moderate diversity): 7-12 responses use the most common approach, with 2-4 responses showing distinct alternative strategies.
- Score 3 (high diversity): ≤6 responses use the same method, with 4+ distinctly different solution strategies present.

**Other Domain Evaluation.**  We conduct additional experiments on MMLU (Hendrycks et al., 2020) and GPQA (Rein et al., 2024) to demonstrate FlowRL's effectiveness extends beyond mathematical reasoning to other domains. We use Qwen-2.5-7B as the base model and follow the math training setup described in Sec 4. As shown in Table 9, FlowRL achieves the highest overall

scores on both benchmarks (72.13% on MMLU and 36.87% on GPQA). These results demonstrate FlowRL's strong generalization capability across different domains beyond the originally tested mathematical reasoning tasks.

Table 8: Human-evaluated diversity scores (Mean $\pm$ Std).

| Method | Score |
|--------|-------|
| R++ | $1.10 \pm 0.20$ |
| GRPO | $1.42 \pm 0.42$ |
| PPO | $1.67 \pm 0.39$ |
| FlowRL | $2.45 \pm 0.35$ |

Table 9: MMLU and GPQA benchmark performance.

| Method | MMLU | GPQA |
|--------|------|------|
| R++ | 71.82 | 27.02 |
| GRPO | 71.87 | 33.08 |
| PPO | 72.10 | 33.84 |
| FlowRL | 72.13 | 36.87 |

## E    EXTENDED RELATED WORK AND COMPARISONS

Recent notable works have addressed similar challenges in large language model reinforcement learning from different perspectives and across various domains. We provide a detailed comparison below to highlight key distinctions and commonalities with existing methods.

**Length Normalization.**    Dr. GRPO (Liu et al., 2025c) proposes an unbiased optimization method that improves token efficiency by removing standard normalization terms from the advantage calculation and removing length terms from the loss objective, while focusing primarily on mathematical reasoning improvements. SRPO (Zhang et al., 2025c) addresses length conflicts through a two-stage training approach (math-first, then coding) and history resampling to filter zero-advantage samples. GSPO (Zheng et al., 2025) conducts gradient analysis and applies length normalization in the sequence-level importance ratio $(s_i(\theta) = (\frac{\pi_\theta(y_i|x)}{\pi_{\theta_{\text{old}}}(y_i|x)})^{\frac{1}{|y_i|}})$ to avoid unstable training, particularly crucial for MoE model training. FlowRL operates as a trajectory-level flow-balance objective that initially faced gradient explosion issues during long CoT reasoning. To overcome this challenge, FlowRL integrates length normalization $(\frac{1}{|y|} \log \pi_\theta(y|x))$ directly into the trajectory balance formulation, ensuring training stability and enabling effective scaling to extended CoT sequences. Unlike approaches requiring domain-specific training strategies, FlowRL's unified formulation naturally handles variable sequence lengths through principled reward shaping within the flow-balance framework, achieving stable optimization across diverse reasoning tasks.

**KL-Related Policy Optimization Methods.**    Kimi-K1.5 (Team et al., 2025) employs on-policy sampling with KL regularization and uses empirical mean of sampled rewards $(\bar{r} = \text{mean}(r(x, y_1, y^*), ..., r(x, y_k, y^*)))$ to approximate the normalizing constant $Z$. This objective has a closed form solution that introduces $\log Z$, where $\gamma$ is a parameter controlling the degree of regularization, maintaining the traditional reward maximization framework. IPO (Azar et al., 2024) addresses overfitting in preference-based learning by using identity mapping $(\Psi = I)$ to maintain effective KL regularization with deterministic preferences, targeting preference-based alignment problems. FlowRL differs by deriving its objective from reverse KL divergence minimization, shifting from reward maximization to reward distribution matching via flow balance. This approach employs a learnable partition function $Z_\phi(x)$ parameterized by a 3-layer MLP and incorporates importance sampling for the entire trajectory balance objective. This approach provides both theoretical rigor through generative flow networks and practical effectiveness across diverse reasoning tasks without requiring preference data or domain-specific training paradigms.

## F    IMPLEMENTATION OF PARTITION FUNCTION $Z_\phi$

We detail the implementation of the partition function $Z_\phi$, covering theoretical foundations and practical aspects.

From the flow perspective: $Z_\phi$ measures the probability flow from the initial state $S_0$. Intuitively, it estimates the denominator—the sum of rewards across all possible paths—enabling conversion to a probability distribution via $\frac{r(\mathbf{x},\mathbf{y})}{Z_\phi(\mathbf{x})}$.

From the implementation perspective: Since the input of $Z_\phi$ corresponds to the initial state, we utilize the prompt representation from the language model. Specifically, we extract the hidden states from the final layer of the language model for all prompt tokens, and compute their mean to obtain a fixed-dimensional representation. This averaged hidden state vector serves as the input feature for computing the scalar partition function value $Z_\phi(\mathbf{x})$.

We conduct comprehensive ablation studies examining: (1) MLP architecture depth (1/3/5 layers); (2) Removing $Z_\phi$ entirely: to quantify how much $Z_\phi$ contributes to the overall performance improvement; (3) Replacing $Z_\phi$ with a constant value: to assess whether adaptivity is necessary or a simple approximation suffices.

The results demonstrate that the learnable partition function $Z_\phi$ is essential for FlowRL's performance. As shown in Table 10, varying MLP depth has minimal impact, with 3-layer MLP performing slightly better. Table 11 shows that removing $Z_\phi$ causes significant drops (-5.62 on AIME 2024, -6.25 on AIME 2025), while using a constant $Z_\phi$ performs even worse (-7.91 and -8.75 respectively). These results confirm that $Z_\phi$ is critical. Theoretically, it is essential for matching the reward distribution.

Table 10: MLP Architecture Depth.

| $Z_\phi$ **Arch.** | **AIME 2024** | **AIME 2025** |
|---|---|---|
| 1-layer MLP | 12.79 | 8.12 |
| 3-layer MLP | 15.41 | 10.83 |
| 5-layer MLP | 10.49 | 6.77 |

Table 11: Partition Function $Z_\phi$.

| **Method** | **AIME 2024** | **AIME 2025** |
|---|---|---|
| FlowRL | 15.41 | 10.83 |
| w/o $Z_\phi$ | 9.79 | 4.58 |
| w/ constant $Z_\phi$ | 7.50 | 2.08 |

## G  TRAINING ANALYSIS

**Training Dynamics.**  We analyze model evolution during training by tracking AIME 2025 accuracy and response length. As shown in Figure 5, FlowRL gradually outperforms GRPO during training.

FlowRL's response length grows faster than GRPO, reaching approximately 2000 tokens by step 100 compared to GRPO's ~1200 tokens. Correspondingly, FlowRL achieves higher AIME 2025 accuracy, with the performance gap widening as training progresses, particularly after step 75 where FlowRL begins to consistently outperform GRPO.

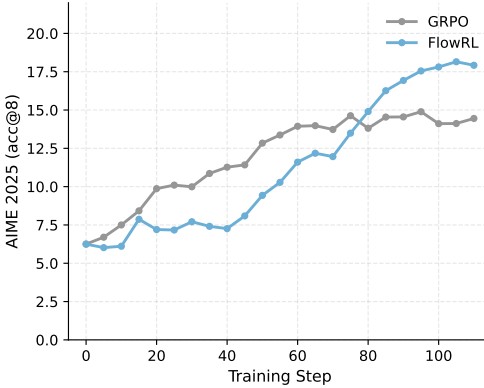 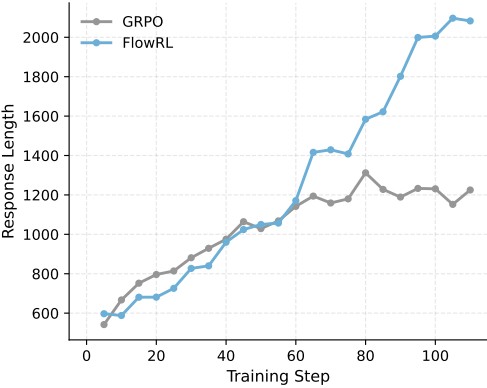

Figure 5: Training dynamics on Qwen2.5-7B, including AIME 2025 Acc@8 (left) and response length (right).

**Reward Distribution Analysis.** We analyze reward distribution statistics during training on Qwen-2.5-32B. FlowRL maintains higher variance than GRPO, indicating exploration of diverse solutions. Specifically, FlowRL achieves higher variance, aligning with flow matching theory that encourages exploration of multiple solution paths.

Table 12: Reward Distribution Statistics.

| Step | GRPO Std | FlowRL Std |
|------|----------|-----------|
| 0    | 0.1087   | 0.1087    |
| 50   | 0.1714   | 0.1341    |
| 100  | 0.0000   | 0.1165    |
| 150  | 0.0323   | 0.1664    |
| 200  | 0.1630   | 0.0730    |
| 245  | 0.0509   | 0.2341    |

**Length Normalization Ablation.** We conduct an ablation study on the length normalization term $(1/|y|)$. The results demonstrate that length normalization is essential for stable training.

Without it, training becomes highly unstable: at step 10, generation length explodes to 1827 tokens with gradient norm spiking to 4.6M; at step 50, length collapses to only 9 tokens, confirming that length normalization is critical for FlowRL's stability.

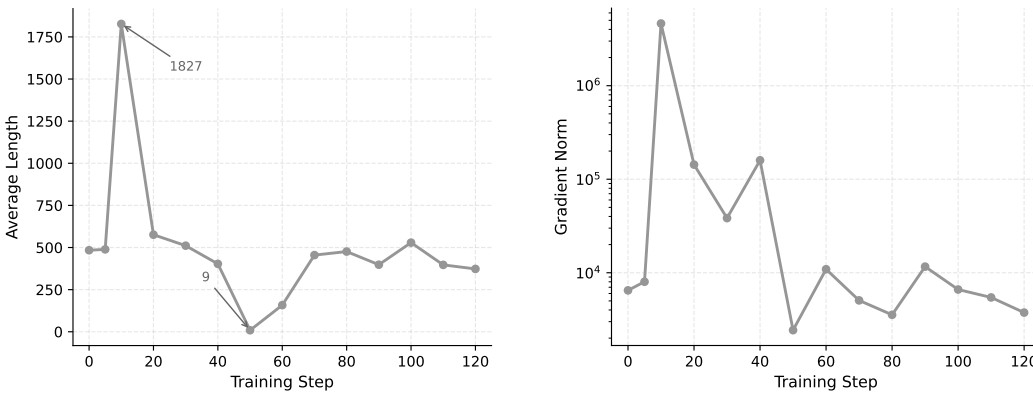

Figure 6: Ablation study on length normalization term $(1/|y|)$. Left: average response length. Right: gradient norm (log scale). Without length normalization, training exhibits severe instability with length explosion/collapse and gradient spikes.

## H    THE USE OF LARGE LANGUAGE MODELS

LLMs (specifically GPT-4o-mini) are used as a judge to evaluate the diversity of solution approaches in our diversity analysis (Figure 4), following Yu et al. (2025a). All core research ideas, theoretical derivations, experimental design, and algorithmic innovations are developed by the authors without LLM assistance. The mathematical formulations and proofs are entirely the work of the human researchers. LLMs do not contribute to the fundamental conceptual development of FlowRL or the core insights about reward distribution matching via flow balance.

## Diversity Evaluation Prompt

**System:** You are evaluating the DIVERSITY of solution approaches for a mathematics competition problem. Focus on detecting even SUBTLE differences in methodology that indicate different problem-solving strategies.

**PROBLEM:**

{problem}

**16 SOLUTION ATTEMPTS:**

{formatted_responses}

**EVALUATION CRITERIA - Rate diversity from 1 to 5:**

**Score 1 - Minimal Diversity:**
- 14+ responses use essentially identical approaches
- Same mathematical setup, same variable choices, same solution path
- Only trivial differences (arithmetic, notation, wording)
- Indicates very low exploration/diversity in the generation process

**Score 2 - Low Diversity:**
- 11-13 responses use the same main approach
- 1-2 alternative approaches appear but are rare
- Minor variations within the dominant method (different substitutions, orderings)
- Some exploration but heavily biased toward one strategy

**Score 3 - Moderate Diversity:**
- 7-10 responses use the most common approach
- 2-3 distinct alternative approaches present
- Noticeable variation in problem setup or mathematical techniques
- Balanced mix showing reasonable exploration

**Score 4 - High Diversity:**
- 4-6 responses use the most common approach
- 3-4 distinct solution strategies well-represented
- Multiple mathematical techniques and problem framings
- Strong evidence of diverse exploration strategies

**Score 5 - Maximum Diversity:**
- No single approach dominates ($\leq 3$ responses use same method)
- 4+ distinctly different solution strategies
- Wide variety of mathematical techniques and creative approaches
- Excellent exploration and generation diversity

**IMPORTANT:** Focus on the DIVERSITY of the attempted approaches. Return ONLY a number from 1 to 5.

