# OpenReview forum: "FlowRL: Matching Reward Distributions for LLM Reasoning"
_ICLR.cc/2026/Conference — ICLR 2026 Poster_

### Official Review · Reviewer_BKo5 · 2025-10-31

**Soundness:** 3
**Presentation:** 3
**Contribution:** 3
**Rating:** 6
**Confidence:** 3

**Summary:**

This paper introduces FlowRL, a novel reinforcement learning algorithm for LLM reasoning that tackles the "mode collapse" problem inherent in traditional reward-maximizing methods like PPO and GRPO. Instead of merely seeking the highest reward, FlowRL's core contribution is to reframe the objective as reward distribution matching, aiming to make the policy's output distribution over reasoning paths proportional to their rewards. Drawing inspiration from GFlowNets, the authors formulate this as a tractable "trajectory balance" objective and introduce crucial technical adaptations—length normalization and a modified form of importance sampling—to make it effective for long-chain-of-thought training. Through strong empirical results on challenging math and code benchmarks, the paper convincingly demonstrates that this new paradigm significantly outperforms strong baselines by promoting greater solution diversity and improving generalization.

**Strengths:**

1. By creatively connecting GFlowNets with LLM reasoning, this paper provides an elegant and novel solution to the well-known problem of mode collapse, moving from the standard reward-maximization objective in RL to a more principled reward distribution matching paradigm.
2. The authors introduce crucial technical solutions—length normalization for stability and modified importance sampling for data efficiency—demonstrating a strong command of the challenges in large-scale RL and bridging the gap between theory and practice.
3. The paper is exceptionally clear, logically motivating and deriving the FlowRL algorithm from first principles. The claims are backed by strong and comprehensive empirical results on challenging math and code benchmarks, where FlowRL consistently and significantly outperforms established baselines like PPO and GRPO.
4. This paper is significant as it introduces a new, effective direction for LLM post-training. By promoting diverse and generalizable reasoning, FlowRL not only pushes state-of-the-art performance but also opens promising avenues for developing more robust and exploratory generative agents, which could have a broad impact on complex problem-solving applications.

**Weaknesses:**

1. The paper only reports final performance metrics, making it impossible to assess whether FlowRL is more or less sample-efficient than GRPO and PPO. Including learning curves (accuracy vs. training steps/samples) is essential to substantiate the method's practical training efficiency, especially given that GFlowNet-based objectives can be sample-inefficient.
2. While the proposed solutions to GFlowNet's instability (e.g., length normalization, detached importance sampling) are empirically effective, their necessity and impact are not deeply ablated, leaving them feeling more like engineering choices than rigorously justified algorithmic components.

**Questions:**

See "Weaknesses".

Additional question:
1. The choice to parameterize the partition function Zφ as a 3-layer MLP is presented without justification. This is a critical architectural decision, yet its impact on performance is unevaluated. What was the rationale for this specific depth, and how sensitive is the model's performance to variations in the Zφ architecture?

---

> ### Author Response · Authors · 2025-11-21
> **Response to Reviewer BKo5 (1/3): Learning Curves (Accuracy vs. Training Steps)**
>
> We deeply appreciate your recognition of our novel GFlowNet-based approach, key technical innovations, and strong empirical results! Below, we provide detailed responses to address your questions.
>
> > [Q1] Including learning curves (accuracy vs. training steps/samples) is essential to substantiate the method's practical training efficiency, especially given that GFlowNet-based objectives can be sample-inefficient.
>
> **We supplement model evolution experiments during training, tracking AIME 2025 accuracy and response length progression. As shown in the table below, FlowRL gradually outperforms GRPO during training.**
>
> FlowRL's response length grows faster than GRPO, reaching approximately 2000 tokens by step 100 compared to GRPO's ~1200 tokens. Correspondingly, FlowRL achieves higher AIME 2025 accuracy, with the performance gap widening as training progresses, particularly after step 75 where FlowRL begins to consistently outperform GRPO.
>
> Table: Training Dynamics on Qwen2.5-7B, including AIME 2025  Acc@8 and response length.
>
> | Step | AIME 2025  |        | Response Length |        |
> | ---- | ----------------- | ------ | --------------- | ------ |
> |      | GRPO              | FlowRL | GRPO            | FlowRL |
> | 0    | 6.25              | 6.25   | -               | -      |
> | 5    | 6.70              | 6.02   | 542             | 597    |
> | 10   | 7.50              | 6.11   | 667             | 588    |
> | 15   | 8.42              | 7.87   | 752             | 681    |
> | 20   | 9.87              | 7.20   | 796             | 681    |
> | 25   | 10.10             | 7.17   | 814             | 726    |
> | 30   | 9.99              | 7.71   | 881             | 827    |
> | 35   | 10.86             | 7.41   | 929             | 840    |
> | 40   | 11.27             | 7.26   | 975             | 960    |
> | 45   | 11.42             | 8.09   | 1064            | 1024   |
> | 50   | 12.84             | 9.43   | 1029            | 1050   |
> | 55   | 13.37             | 10.28  | 1068            | 1057   |
> | 60   | 13.94             | 11.60  | 1142            | 1171   |
> | 65   | 13.98             | 12.18  | 1194            | 1416   |
> | 70   | 13.73             | 11.96  | 1159            | 1429   |
> | 75   | 14.63             | 13.49  | 1180            | 1408   |
> | 80   | 13.81             | 14.90  | 1312            | 1584   |
> | 85   | 14.54             | 16.26  | 1228            | 1622   |
> | 90   | 14.55             | 16.93  | 1189            | 1802   |
> | 95   | 14.89             | 17.55  | 1233            | 1999   |
> | 100  | 14.11             | 17.81  | 1231            | 2006   |
> | 105  | 14.12             | 18.15  | 1152            | 2097   |
> | 110  | 14.45             | 17.92  | 1225            | 2083   |
> | 115  | 15.00             | 18.77  | 1305            | 2440   |
> | 120  | 15.26             | 18.70  | 1279            | 2221   |
> | 125  | 15.48             | 19.49  | 1253            | 1838   |

---

> ### Author Response · Authors · 2025-11-21
> **Response to Reviewer BKo5 (2/3): Ablations on Length Normalization**
>
> > [Q2] While the proposed solutions to GFlowNet's instability (e.g., length normalization, detached importance sampling) are empirically effective, their necessity and impact are not deeply ablated, leaving them feeling more like engineering choices than rigorously justified algorithmic components.
>
> Thank you for the suggestion. We have already presented ablation results for importance sampling in Table 3 of the original paper. Below, **we provide an ablation study on the length normalization term (1/|y|), demonstrating that length normalization is essential for stable training.**
>
> Without it, training becomes highly unstable: at step 10, generation length explodes to 1827 tokens with gradient norm spiking to 4.6M; at step 50, length collapses to only 9 tokens. Overall, 25.4% of training steps experience gradient explosions (>50k), confirming that length normalization is critical for FlowRL's stability.
>
> Table: Ablation Study on Length Normalization Term (1/|y|).
>
> | Step | Training Reward | Avg Length | Grad Norm |
> | ---- | --------------- | ---------- | --------- |
> | 0    | 0.3167          | 484        | 6482.0    |
> | 5    | 0.3630          | 489        | 8014.6    |
> | 10   | 0.0090          | 1827       | 4623571.5 |
> | 20   | 0.4568          | 576        | 143240.3  |
> | 30   | 0.4648          | 511        | 38394.3   |
> | 40   | 0.3223          | 403        | 159334.6  |
> | 50   | 0.0574          | 9          | 2432.7    |
> | 60   | 0.4106          | 158        | 10903.6   |
> | 70   | 0.5276          | 455        | 5063.7    |
> | 80   | 0.5439          | 476        | 3550.1    |
> | 90   | 0.3938          | 398        | 11625.1   |
> | 100  | 0.5894          | 529        | 6618.9    |
> | 110  | 0.5449          | 397        | 5444.7    |
> | 120  | 0.5051          | 373        | 3751.2    |

---

> ### Author Response · Authors · 2025-11-23
> **Response to Reviewer BKo5 (3/3): Design of Partition Function $Z_\phi$**
>
> > [Q3] The choice to parameterize the partition function $Z_\phi$ as a 3-layer MLP is presented without justification.  What was the rationale for this specific depth, and how sensitive is the model's performance to variations in the Zφ architecture?
>
> As stated in lines 263–265 of our original paper, **we follow [1] in using a randomly initialized 3-layer MLP.**
>
> To further address your concerns, **we conduct comprehensive ablation studies** examining: (1) MLP architecture depth (1/3/5 layers); (2) Removing $Z_\phi$ entirely: to quantify how much $Z_\phi$ contributes to the overall performance improvement; (3) Replacing $Z_\phi$ with a constant value: to assess whether adaptivity is necessary or a simple approximation suffices [2, 3].
>
> **The results demonstrate that the learnable partition function $Z_\phi$ is essential for FlowRL's performance.**
>
> As shown in Table 1, varying MLP depth has minimal impact, with 3-layer MLP performing slightly better. Table 2 shows that removing $Z_\phi$ causes significant drops (-5.62 on AIME 2024, -6.25 on AIME 2025), while using a constant $Z_\phi$ performs even worse (-7.91 and -8.75 respectively). These results confirm that $Z_\phi$ is critical—theoretically, it is essential for matching the reward distribution.
>
> Table 1: Ablation Study on $Z_\phi$ MLP Architecture Depth
>
> | $Z_\phi$ Architecture | AIME 2024 | AIME 2025 |
> | --------------------- | --------- | --------- |
> | 1-layer MLP           | 12.79     | 8.12      |
> | 3-layer MLP           | 15.41     | 10.83     |
> | 5-layer MLP           | 10.49     | 6.77      |
>
> Table 2: Ablation Study on Partition Function $Z_\phi$
>
> | Method                   | AIME 2024 | AIME 2025 |
> | ------------------------ | --------- | --------- |
> | FlowRL                   | 15.41     | 10.83     |
> | (1) w/o $Z_\phi$         | 9.79      | 4.58      |
> | (2) w/ constant $Z_\phi$ | 7.50      | 2.08      |
>
> References:
>
> [1] Lee S, Kim M, Cherif L, et al. Learning diverse attacks on large language models for robust red-teaming and safety tuning. ICLR 2025.
>
> [1] Yu F, Jiang L, Kang H, et al. Flow of Reasoning: Training LLMs for Divergent Reasoning with Minimal Examples. ICML 2025.
>
> [2] Zhang D W, Rainone C, Peschl M, et al. Robust scheduling with gflownets. ICLR 2023.

---

### Official Review · Reviewer_Ngao · 2025-10-31

**Soundness:** 3
**Presentation:** 3
**Contribution:** 3
**Rating:** 6
**Confidence:** 3

**Summary:**

1. FlowRL shifts RL from maximizing scalar rewards to matching the full reward distribution, eliminating mode collapse and enriching reasoning diversity.
2. A learnable partition-function network approximates the intractable normalization constant. It also employs length normalization and importance sampling to enhance training stability.
3. Experiments demonstrate that the proposed method outperforms both PPO and GRPO on math and code reasoning tasks.

**Strengths:**

1. The insight is highly meaningful—it enhances the representational capacity of rewards and constitutes a valuable innovative attempt.
2. Supported by solid theoretical justifications and in-depth analysis.
3. The paper is well-written, fluent, and easy to follow.

**Weaknesses:**

1. Please refer to the "Questions" section for details.

**Questions:**

1. The ablation study omits an ablation on “reward shaping via length normalization.”
2. Why a 3-layer MLP is chosen to parameterize Z_ϕ could be further investigated and compared with alternative designs.
3. A statistical analysis of reward distributions under different training scenarios—detailing their shapes, similarities, and differences—would be valuable.

---

> ### Author Response · Authors · 2025-11-21
> **Response to Reviewer Ngao (1/3): Ablations on Reward Shaping via Length Normalization.**
>
> Thank you for appreciating the meaningful insight of our approach, its solid theoretical foundation with in-depth analysis, and the clarity of our writing!
>
> > [Q1] The ablation study omits an ablation on “reward shaping via length normalization”.
>
> **We add an ablation study on the length normalization term (1/|y|). The results demonstrate that length normalization is essential for stable training.**
>
> Without it, training becomes highly unstable: at step 10, generation length explodes to 1827 tokens with gradient norm spiking to 4.6M; at step 50, length collapses to only 9 tokens. Overall, 25.4% of training steps experience gradient explosions (>50k), confirming that length normalization is critical for FlowRL's stability.
>
> Table: Ablation Study on Length Normalization Term (1/|y|).
>
> | Step | Training Reward | Avg Length | Grad Norm |
> | ---- | --------------- | ---------- | --------- |
> | 0    | 0.3167          | 484        | 6482.0    |
> | 5    | 0.3630          | 489        | 8014.6    |
> | 10   | 0.0090          | 1827       | 4623571.5 |
> | 20   | 0.4568          | 576        | 143240.3  |
> | 30   | 0.4648          | 511        | 38394.3   |
> | 40   | 0.3223          | 403        | 159334.6  |
> | 50   | 0.0574          | 9          | 2432.7    |
> | 60   | 0.4106          | 158        | 10903.6   |
> | 70   | 0.5276          | 455        | 5063.7    |
> | 80   | 0.5439          | 476        | 3550.1    |
> | 90   | 0.3938          | 398        | 11625.1   |
> | 100  | 0.5894          | 529        | 6618.9    |
> | 110  | 0.5449          | 397        | 5444.7    |
> | 120  | 0.5051          | 373        | 3751.2    |

---

> ### Author Response · Authors · 2025-11-23
> **Response to Reviewer Ngao (2/3): Analysis of 3-layer MLP**
>
> > [Q2] Why a 3-layer MLP is chosen to parameterize Z_ϕ could be further investigated and compared with alternative designs.
>
> As stated in lines 263–265 of our original paper, **we follow [1] in using a randomly initialized 3-layer MLP.**
>
> To further address your concerns, **we add comprehensive ablation studies** examining: (1) MLP architecture depth (1/3/5 layers); (2) Removing $Z_\phi$ entirely: to quantify how much $Z_\phi$ contributes to the overall performance improvement; (3) Replacing $Z_\phi$ with a constant value: to assess whether adaptivity is necessary or a simple approximation suffices [2, 3].
>
> **The results demonstrate that the learnable partition function $Z_\phi$ is essential for FlowRL's performance.**
>
> As shown in Table 1, varying MLP depth has minimal impact, with 3-layer MLP performing slightly better. Table 2 shows that removing $Z_\phi$ causes significant drops, while using a constant $Z_\phi$ performs even worse. These results confirm that $Z_\phi$ is critical. Theoretically, it is essential for matching the reward distribution.
>
> Table 1: Ablation Study on $Z_\phi$ MLP Architecture Depth
>
> | $Z_\phi$ Architecture | AIME 2024 | AIME 2025 |
> | --------------------- | --------- | --------- |
> | 1-layer MLP           | 12.79     | 8.12      |
> | **3-layer MLP**           | **15.41**     | **10.83**     |
> | 5-layer MLP           | 10.49     | 6.77      |
>
> Table 2: Ablation Study on Partition Function $Z_\phi$
>
> | Method                   | AIME 2024 | AIME 2025 |
> | ------------------------ | --------- | --------- |
> | **FlowRL**                   | **15.41**     | **10.83**     |
> | (1) w/o $Z_\phi$         | 9.79      | 4.58      |
> | (2) w/ constant $Z_\phi$ | 7.50      | 2.08      |
>
> References:
>
> [1] Lee S, Kim M, Cherif L, et al. Learning diverse attacks on large language models for robust red-teaming and safety tuning. ICLR 2025.
>
> [2] Yu F, Jiang L, Kang H, et al. Flow of Reasoning: Training LLMs for Divergent Reasoning with Minimal Examples. ICML 2025.
>
> [3] Zhang D W, Rainone C, Peschl M, et al. Robust scheduling with gflownets. ICLR 2023.

---

> > ### Author Response · Authors · 2025-11-23
> > **Response to Reviewer Ngao (3/3): Statistical Analysis of Reward Distributions**
> >
> > > [Q3] A statistical analysis of reward distributions under different training scenarios—detailing their shapes, similarities, and differences—would be valuable.
> >
> > We analyze reward distribution statistics during training on Qwen-2.5-32B. **FlowRL maintains higher variance than GRPO, indicating exploration of diverse solutions.** Specifically, FlowRL achieves higher variance, aligning with flow matching theory that encourages exploration of multiple solution paths.
> >
> > Table: Reward Distribution Statistics
> >
> > | Step | GRPO STD | FlowRL STD |
> > | ---- | -------- | ---------- |
> > | 0    | 0.1087   | 0.1087     |
> > | 50   | 0.1714   | 0.1341     |
> > | 100  | 0.0000   | 0.1165     |
> > | 150  | 0.0323   | 0.1664     |
> > | 200  | 0.1630   | 0.0730     |
> > | 245  | 0.0509   | 0.2341     |

---

### Official Review · Reviewer_oDPD · 2025-10-31

**Soundness:** 3
**Presentation:** 3
**Contribution:** 2
**Rating:** 6
**Confidence:** 3

**Summary:**

FlowRL introduces a new RL objective for LLM post training that is motivated by connecting reverse KL with GFlowNet. In this new formulation, the advantage is regularized by the important sampling term and a learned partition function (via a small MLP from the mean pooled LLM hidden state) so that the model's output diversity is preserved. The author demonstrates effectiveness of the proposed method on Qwen-2.5 and DeepSeek-R1-Distilled Qwen models on math and reasoning benchmarks.

**Strengths:**

The paper is generally well written and easy to follow.

The biggest contribution from the paper is, in my opinion, the formulation of using GFlowNets to implement reserve KL for distribution matching training objective. The idea is elegant and the ablations seem to cover important tricks that are necessary for implementation on models at scale. It is great to see works that are both theoretically grounded and implementationally sound.

**Weaknesses:**

Given my lack of familiarity of the GFlowNet lineage of work, I can only comment on the algorithm from a purely pragmatic perspective.

The main drawback is that the FlowRL's benefits mainly come from important sampling to control model entropy, which could be achieved more easily with other methods without additional learnable parameters (in the partition function). In particular, the objective has 2 main components that are augumenting the advantage function: 1) Importance sampling, 2) partition function. From the ablations in the work, it appears that the IS part of the augmentation is very important, however, it is not clear whether the partition function is needed at all in practice. _If_ partition function is not necessary from a final model performance perspective, then FlowRL is similar to GRPO with a penalty on pi_theta/pi_ref, which in spirit looks quite similar to a KL penalty. Therefore, from a pragmatic standpoint, the question is whether the complexity of FlowRL necessary for the performance gain that we observe.

**Questions:**

It would be really interesting to ablate Z_phi to see how much partition function is helping. In principle this should help reducing training variance, but then the question is what if we simply tuning other hparams like number of rollouts/sample.

It would also be interesting to see how the model evolves during training from things like AIME vs. generation length, or AIME vs. training flops. It is not immediately clear that the gains reported in FlowRL would scale with model size and training flops.

---

> ### Author Response · Authors · 2025-11-21
> **Response to Reviewer oDPD (1/2): Ablations on Partition Function $Z_\phi$**
>
> Thank you for recognizing our clarity, elegant GFlowNet formulation for reverse KL, and theoretical soundness! Please see our detailed response to your concerns below.
>
>
> > [Weaknesses & Q1] It would be really interesting to ablate Z_phi to see how much partition function is helping.
>
> **We add ablation studies by removing $Z_\phi $ and using a constant $Z_\phi $. The results demonstrate that the learnable partition function is essential for FlowRL's performance.**
>
> Specifically, we conduct two ablation experiments $Z_\phi$ : (1) directly removing $Z_\phi$ as suggested, and (2) replacing the learned $Z_\phi$ with a constant value computed as the average reward within each batch, inspired by [1, 2].  As shown in the table below, removing $Z_\phi $ causes significant drops on AIME 2024  and AIME 2025. Using a constant $Z_\phi $ performs even worse, demonstrating that a static approximation cannot replace the adaptive partition function.
>
> These results confirm the critical importance of  $Z_\phi$. From a theoretical perspective, the partition function is an essential component for matching the reward distribution.
>
> Table: Ablation Study on Partition Function $Z_\phi$
>
> | Method                   | AIME 2024 | AIME 2025 |
> | ------------------------ | --------- | --------- |
> | FlowRL                   | 15.41     | 10.83     |
> | (1) w/o $Z_\phi$         | 9.79      | 4.58      |
> | (2) w/ constant $Z_\phi$ | 7.50      | 2.08      |
>
> References:
>
> [1] Yu F, Jiang L, Kang H, et al. Flow of Reasoning: Training LLMs for Divergent Reasoning with Minimal Examples. ICML 2025.
>
> [2] Zhang D W, Rainone C, Peschl M, et al. Robust scheduling with gflownets. ICLR 2023.

---

> > ### Author Response · Authors · 2025-11-23
> > **Response to Reviewer oDPD (2/2): Evolution of Model Performance during Training**
> >
> > >  [Q2] It would also be interesting to see how the model evolves during training from things like AIME vs. generation length, or AIME vs. training flops.
> >
> > **We supplement model evolution experiments during training, tracking AIME 2025 accuracy and response length progression. As shown in the table below, FlowRL gradually outperforms GRPO during training.**
> >
> > FlowRL's response length grows faster than GRPO, reaching approximately 2000 tokens by step 100 compared to GRPO's ~1200 tokens.
> >
> > Table: Training Dynamics on Qwen2.5-7B, including AIME 2025  Acc@8 and response length.
> >
> > | Step | AIME 2025 |        | Response Length |        |
> > | ---- | ----------------- | ------ | --------------- | ------ |
> > |      | GRPO              | FlowRL | GRPO            | FlowRL |
> > | 0    | 6.25              | 6.25   | -               | -      |
> > | 5    | 6.70              | 6.02   | 542             | 597    |
> > | 10   | 7.50              | 6.11   | 667             | 588    |
> > | 15   | 8.42              | 7.87   | 752             | 681    |
> > | 20   | 9.87              | 7.20   | 796             | 681    |
> > | 25   | 10.10             | 7.17   | 814             | 726    |
> > | 30   | 9.99              | 7.71   | 881             | 827    |
> > | 35   | 10.86             | 7.41   | 929             | 840    |
> > | 40   | 11.27             | 7.26   | 975             | 960    |
> > | 45   | 11.42             | 8.09   | 1064            | 1024   |
> > | 50   | 12.84             | 9.43   | 1029            | 1050   |
> > | 55   | 13.37             | 10.28  | 1068            | 1057   |
> > | 60   | 13.94             | 11.60  | 1142            | 1171   |
> > | 65   | 13.98             | 12.18  | 1194            | 1416   |
> > | 70   | 13.73             | 11.96  | 1159            | 1429   |
> > | 75   | 14.63             | 13.49  | 1180            | 1408   |
> > | 80   | 13.81             | 14.90  | 1312            | 1584   |
> > | 85   | 14.54             | 16.26  | 1228            | 1622   |
> > | 90   | 14.55             | 16.93  | 1189            | 1802   |
> > | 95   | 14.89             | 17.55  | 1233            | 1999   |
> > | 100  | 14.11             | 17.81  | 1231            | 2006   |
> > | 105  | 14.12             | 18.15  | 1152            | 2097   |
> > | 110  | 14.45             | 17.92  | 1225            | 2083   |
> > | 115  | 15.00             | 18.77  | 1305            | 2440   |
> > | 120  | 15.26             | 18.70  | 1279            | 2221   |
> > | 125  | 15.48             | 19.49  | 1253            | 1838   |

---

> > > ### Comment · Reviewer_oDPD · 2025-11-23
> > >
> > > Thank you for the additional information.
> > >
> > > Looking at this updated results, FlowRL seems to have
> > >
> > > 1. better final performance
> > > 2. worse token efficiency (if you plot AIME vs. length)
> > >
> > > In my own experience, for reasoning tasks such as AIME, the model's performance typically increase with generation length , which is what we observe in the result above. Consequently, when comparing methods/models, the performance given generation length is, in my opinion, more indicative of the effectiveness of a given method. From this perspective, FlowRL seems to be unfortunately worse than the GRPO baseline.
> > >
> > > I won't hold this against the authors since it doesn't change the original assessment in my review. However, it would be great if the authors could explore if FlowRL's stronger length growth is a result of the algorithmic design or an artifact of the hyperparamters/base policy/dataset. It would be of great value to the community if we have a better understanding and thus control of the length behavior of LLMs during RL.
> > >
> > > All in all, thank you for the response. I maintain my original score.

---

> ### Author Response · Authors · 2025-11-23
> **Analysis of Stronger Length Growth & Discussion of token efficiency**
>
> Thank you for your prompt and thoughtful response.
>
> **We agree that a longer CoT is a key factor contributing to FlowRL's performance advantage, and we consider this length increase to be a strength rather than a disadvantage of FlowRL.**
>
> **Scaling up to 32K, FlowRL vs. CISPO [1]:** To further investigate FlowRL's length growth characteristic, we conduct additional scaling experiments. As shown in the following tables, compared to CISPO, which has been considered a strong RL method when scaling RL compute [2], FlowRL progressively generates longer responses throughout training, which validates our goal of encouraging more exploration. And also confirming that: **FlowRL's length advantage is inherent to the algorithm itself, not an artifact of hyperparameters.**
>
> -----
>
> **Discussion of token efficiency**:  in industrial-scale deployment [3], we prioritize achieving strong performance first—preferring longer CoT to obtain better results. Token efficiency can then be improved through mechanisms like "thinking mode" switching (as in Qwen [4]) to shorten CoT when appropriate, since efficiency is meaningful only when performance is ensured.  Our current approach for industrial-level deployment and scaling is to conduct additional fine-tuning for efficient inference while maintaining strong performance.
>
> BTW, we warmly welcome more of your insights and suggestions. We are actively conducting future work on this topic and would be delighted to discuss this further with you if you are interested.
>
> ------
>
> **Settings: 64 H200 GPUs, max response length 32k, 7B base model.**
>
> Note that: Due to limited GPU availability, validation results will be included in the final paper once experiments are completed.
>
> Table 1: Overall comparison of response lengths between FlowRL and CISPO.
>
> | Algorithm  | Total Steps | Avg Response Length | Min Length | Max Length |
> | ---------- | ----------- | ------------------- | ---------- | ---------- |
> | **FlowRL** | 90          | **19,418**          | 17,910     | 21,286     |
> | **CISPO**  | 90          | **16,802**          | 14,590     | 19,772     |
>
> Table 2: Detailed training dynamics showing response length at 5-step intervals for FlowRL and CISPO.
>
> | STEP | RESPONSE LENGTH |            |
> | ---- | --------------- | ---------- |
> |      | **CISPO**       | **FlowRL** |
> | 0    | --              | --         |
> | 5    | 18,270          | 18,640     |
> | 10   | 16,337          | 19,773     |
> | 15   | 15,892          | 20,043     |
> | 20   | 14,609          | 21,286     |
> | 25   | 15,941          | 20,320     |
> | 30   | 16,537          | 19,813     |
> | 35   | 16,757          | 19,665     |
> | 40   | 16,135          | 19,406     |
> | 45   | 15,012          | 19,906     |
> | 50   | 17,173          | 19,100     |
> | 55   | 16,392          | 19,078     |
> | 60   | 18,742          | 18,662     |
> | 65   | 16,299          | 19,159     |
> | 70   | 16,077          | 18,803     |
> | 75   | 18,420          | 19,027     |
> | 80   | 15,749          | 19,442     |
> | 85   | 16,493          | 18,345     |
> | 90   | 16,848          | 18,512     |
>
>
> [1] MiniMax-M1: Scaling Test-Time Compute Efficiently with Lightning Attention.
>
> [2] The Art of Scaling Reinforcement Learning Compute for LLMs.
>
> [3] DeepSeek-R1: Incentivizing reasoning capability in LLMs via reinforcement learning.
>
> [4] Qwen3 technical report.

---

### Official Review · Reviewer_vbie · 2025-11-03

**Soundness:** 4
**Presentation:** 4
**Contribution:** 4
**Rating:** 8
**Confidence:** 2

**Summary:**

This paper proposes FlowRL, a reinforcement learning algorithm that matches the full reward distribution via flow balancing instead of solely maximizing rewards for LLM post-training. Unlike reward-maximizing methods (REINFORCE++, PPO, GRPO) that tend to over-optimize dominant modes while neglecting less frequent but valid reasoning paths, FlowRL transforms scalar rewards into a normalized target distribution using a learnable partition function Zφ(x), then minimizes the reverse KL divergence between the policy and this distribution.The authors prove gradient equivalence between this KL objective and the trajectory balance formulation from GFlowNets, and introduce length normalization and importance sampling to address gradient explosion and sampling mismatch in long CoT reasoning.

**Strengths:**

- Novel application of GFlowNets to LLM reasoning through distribution-matching framework
- Rigorous mathematical formulation with gradient equivalence proof
- Exceptionally clear presentation with intuitive figures
- Achieves substantial improvements

**Weaknesses:**

- No human evaluation: All diversity assessments rely on GPT-4o-mini judgments
- Only tested on reasoning tasks (math, code); unclear if FlowRL works for other domains
- All experiments use thinking models or reasoning-focused tasks; what about non-reasoning tasks?

**Questions:**

As mentioned in Weaknesses.

---

> ### Author Response · Authors · 2025-11-21
> **Response to Reviewer vbie (1/2): Human Evaluation**
>
> We sincerely thank you for your positive feedback on our novelty, theoretical proof, and significant improvements!
> We hope the following responses address your concerns.
>
> > [Q1] No human evaluation: All diversity assessments rely on GPT-4o-mini judgments
>
> **We add a comprehensive human evaluation that demonstrates strong agreement with GPT-4o-mini assessments.** Both evaluators independently identify FlowRL as the most diverse method and R++ as the least diverse, with GRPO and PPO showing intermediate diversity levels.
>
> Human Instruction:  As a human evaluator, assess the diversity of solutions for each problem by examining 16 solution attempts per method. Rate diversity on a 1-3 scale based on the following criteria:
>
> - Score 1 (Low Diversity): 13+ responses use essentially identical approaches with only trivial differences in arithmetic, notation, or wording.
>
> - Score 2 (Moderate Diversity): 7-12 responses use the most common approach, with 2-4 responses showing distinct alternative strategies.
>
> - Score 3 (High Diversity): ≤6 responses use the same method, with 4+ distinctly different solution strategies present.
>
> Table: Human-Evaluated Diversity Scores (Mean ± Std)
>
> | Method | Human Evaluation (Mean ± Std) |
> | ------ | ------------------------ |
> | R++    | 1.10 ± 0.20              |
> | GRPO   | 1.42 ± 0.42              |
> | PPO    | 1.67 ± 0.39              |
> | **FlowRL** | **2.45 ± 0.35**              |

---

> > ### Author Response · Authors · 2025-11-23
> > **Response to Reviewer vbie (2/2): Generalization to Other Domains and Non-Reasoning Tasks**
> >
> > > [Q2 & Q3]  unclear if FlowRL works for other domains,  what about non-reasoning tasks?
> >
> > **We add experiments on MMLU and GPQA to demonstrate FlowRL's effectiveness extends beyond mathematical reasoning to other domains.**
> >
> > As shown in table below, FlowRL achieves the highest overall scores on both benchmarks (72.13% on MMLU and 36.87% on GPQA).  These results demonstrate FlowRL's strong generalization capability across different domains beyond the originally tested mathematical reasoning tasks.
> >
> > Table 1: MMLU Benchmark Performance. We use Qwen-2.5-7B as the base model and follow the training setup described in Section 4 of the original paper.
> >
> > | METHOD | MMLU  | GPQA  |
> > | ------ | ----- | ----- |
> > | R++    | 71.82 | 27.02 |
> > | GRPO   | 71.87 | 33.08 |
> > | PPO    | 72.10 | 33.84 |
> > | **FlowRL** | **72.13** | **36.87** |

---

### Author Response · Authors · 2025-11-27
**Summary of Rebuttal**

Dear Reviewers,

Thank you for your valuable comments. We have revised our manuscript and conducted additional experiments based on your feedback. Changes are highlighted in blue. Summary of modifications:

1. **Human Evaluation** (@Reviewer vbie): We conduct human evaluation on AIME 24/25 rollouts. Results in Table 8 confirm FlowRL as the most diverse method, with strong human-GPT agreement.

2. **More Domains Evaluation** (@Reviewer vbie): We evaluate on MMLU and GPQA. As shown in Table 9, FlowRL achieves the highest scores, demonstrating cross-domain generalization.

3. **Ablation Study on $Z_\phi$** (@Reviewer oDPD, Ngao and BKo5): We ablate MLP depth and partition function $Z_\phi$. Results in Table 10 and 11 confirm $Z_\phi$ is critical for performance.

4. **Training Dynamics** (@Reviewer oDPD and BKo5): We track AIME 2025 accuracy and response length during training. As shown in Figure 5, FlowRL gradually outperforms GRPO.

5. **Reward Distribution Analysis** (@Reviewer Ngao): FlowRL maintains higher reward variance than GRPO, supporting flow matching theory's exploration of diverse solutions (Table 12).

6. **Length Normalization Ablation** (@Reviewer Ngao and BKo5): We show length normalization ($1/|y|$) is essential for stable training, preventing gradient explosions and length collapse (Figure 6).

We sincerely thank all reviewers and ACs for their feedback.

---

### Meta-Review · Area_Chair_7izq · 2026-01-22

**Summary:**

This paper proposes FlowRL, a reinforcement learning objective for LLM post-training that shifts from reward maximization to reward distribution matching via reverse KL and GFlowNet-style trajectory balance. Reviewers found the formulation theoretically sound, clearly presented, and empirically strong, with consistent gains over PPO/GRPO on math and code reasoning benchmarks.

**Reviewer Concerns:**

Addressed:
* Added human evaluation confirms GPT-based diversity judgments.

* Added broader-domain results (MMLU, GPQA) support generalization beyond math/code.

* Partition function $Z_{\phi}$ ablations show it is critical; constant or removed variants degrade performance.

* Length normalization ablation demonstrates necessity for stability.

* Training dynamics and learning curves were added.

Partially outstanding:

* FlowRL tends to generate longer responses, and performance may correlate with length. Comparisons under matched token budgets would further strengthen the claims, but this is not a blocker.

**Reviewer Scores:**

All reviewers would likely keep their original scores. No reviewer raised new blocking concerns after the rebuttal.

---

### Decision · Program_Chairs · 2026-01-26

Accept (Poster)